# Over-parameterized Adversarial Training: An Analysis Overcoming the Curse of Dimensionality

**Yi Zhang\***
Princeton University
y.zhang@cs.princeton.edu

**Orestis Plevrakis**$^*$
Princeton University
orestisp@cs.princeton.edu

**Simon S. Du**
University of Washington
ssdu@cs.washington.edu

**Xingguo Li**
Princeton University
xingguol@cs.princeton.edu

**Zhao Song**
Institute of Advanced Study
zhao@ias.edu

**Sanjeev Arora**
Princeton University and IAS
arora@cs.princeton.edu

## Abstract

Adversarial training is a popular method to give neural nets robustness against adversarial perturbations. In practice adversarial training leads to low robust training loss. However, a rigorous explanation for why this happens under natural conditions is still missing. Recently a convergence theory for standard (non-adversarial) training was developed by various groups for *very over-parametrized* nets. It is unclear how to extend these results to adversarial training because of the min-max objective. Recently, a first step towards this direction was made by [14] using tools from online learning, but they require the width of the net and the running time to be *exponential* in input dimension $d$, and they consider an activation function that is not used in practice. Our work proves convergence to low robust training loss for *polynomial* width and running time, instead of exponential, under natural assumptions and with ReLU activation. Key element of our proof is showing that ReLU networks near initialization can approximate the step function, which may be of independent interest.

## 1 Introduction

Deep neural networks trained by gradient based methods tend to change their answer (incorrectly) after small adversarial perturbations in inputs [25]. Much effort has been spent to make deep nets resistant to such perturbations but adversarial training with a natural min-max objective [20] stands out as one of the most effective approaches according to [9, 7].

One interpretation of the min-max formulation is a certain two-player game between a neural network learner and an adversary who is allowed to perturb the input within certain constraints. In each round, the adversary generates new adversarial examples against the current network, on which the learner takes a gradient step to decrease its prediction loss in response (see Algorithm 1).

It is empirically observed that, when the neural network is initialized randomly, this training algorithm is efficient and computes a reasonably sized neural net that is robust on (at least) the *training* examples [20]. We're interested in theoretical understanding of this phenomenon: *Why does adversarial*

---

$^*$Equal contribution

*training efficiently find a feasibly sized neural net to fit training data robustly?* In recent years, a convergence theory has been developed for non-adversarial training: it explains the ability of gradient descent to achieve small training loss, provided the neural nets are fairly *over-parametrized*. But it is quite unclear whether similar analysis can be applied to adversarial training setting where the inputs are perturbed. Furthermore, while the algorithm is reminiscent of well-studied no-regret dynamics for finding equilibria in two-player zero-sum convex/concave games [17], here the game value is training loss, and hence non-convex. Thus it is unclear if training leads to small robust training loss.

A study of such issues was initiated in [14]. For two-layer nets with *quadratic ReLU* activation[2] they were able to show that if input is in $\mathbb{R}^d$ then training can achieve robust loss at most $\epsilon$ provided the net's width is $(1/\epsilon)^{\Omega(d)}$ (the number of required iterations is also that large)[3]. This is very extreme over-parametrization, and this *curse of dimensionality* is inherent to their argument. They left as an open problem the possibility to improve the width requirement, which is the theme of our paper.

**Our contributions:** Under a standard and natural assumption that training data are well-separated with respect to the magnitude of the adversarial perturbations (also verified for popular datasets in Figure 1) we show the following:

- That there exists a two-layer ReLU neural network with width $\mathrm{poly}\,(d, n/\epsilon)$ near Gaussian random initialization that achieves $\epsilon$ robust training loss.
- That starting from Gaussian random initialization, standard adversarial training (Algorithm 1) converges to such a network in $\mathrm{poly}\,(d, n/\epsilon)$ iterations.
- New result in approximation theory, specifically the existence of a good approximation to the step function by a polynomially wide two-layer ReLU network with weights close to the standard gaussian initialization. Such approximation result may be of further use in the emerging theory of over-parameterized nets.

**Paper structure.** This paper is organized as follows. In section 2, we give an overview of the related works. In section 3, we present our notation, the adversarial training algorithm, the separability condition and we argue why the training examples being well-separated is a natural assumption. In section 4, we formally state our main result and in section 5 we give an overview of its proof. In section 6 we elaborate more on the core part of the proof, which is the existence of a net close to initialization that robustly fits the training data.

## 2   Related Works

**Adversarial examples and defense.** The seminal paper [25] discovered the existence of adversarial examples. Since its discovery, numerous defense methods have been proposed to make neural nets robust to perturbations constrained in a ball with respect to a certain norm (e.g. $\ell_2$, $\ell_\infty$). These methods span an extremely wide spectrum including certification [22, 26], input transformation [8, 15], randomization [27], adversarial training [20], etc. Recent studies on evaluating the effectiveness of the aforementioned defenses by [9, 7] reveals that adversarial training dominates the others. One empirical observation made in [20] is that adversarial training can always make wide nets achieve small robust training loss.

**Convergence via over-parameterization.** Recently, there has been a tremendous progress in understanding the "small training loss" phenomenon in standard (non-adversarial) training [19, 11, 4, 3, 10, 6, 5, 24, 28, 21]. A convergence theory has been developed to show that, when randomly initialized, gradient descent and stochastic gradient descent converge to small training loss in polynomially many iterations when the network has polynomial width in terms of the number of training examples. These papers studied over-paramterized neural networks in the neural tangent kernel (NTK) regime [18].

**Convergence of adversarial training.** There is a growing interest in analyzing convergence properties of adversarial training. [14] made a first attempt towards extending the aforementioned results in

standard training to adversarial training. Like previous works on the convergence of (non-adversasrial) gradient descent for over-parameterized neural networks, this work also considered the NTK regime. First of all, they prove that adversarial training with an artificial projection step always finds a multi-layer ReLU net that is $\epsilon$-optimal within the neighborhood near initialization, but the optimal robust loss could be large. Secondly, for two-layer quadratic ReLU net, they managed to prove that small adversarial loss will be achieved, but crucially the required width and running time are $(1/\epsilon)^{\Omega(d)}$. Their argument suffers *the curse of dimensionality*, because it relies on the universality of the induced Reproducing Kernel Hilbert Space followed by a random feature approximation. In contrast, we take a closer look on how to approximate a robust classifier with ReLU networks near their initialization using techniques from polynomial approximation and manage to overcome this problem. In addition, our convergence analysis applies to ReLU activated nets without additional projection steps.

**Polynomial approximation.** A key technique in our proof is a polynomial approximation to the step function on interval $[-1, -\eta] \cup [\eta, 1]$ which has been an important subject [1, 13, 12]. For $\epsilon$-uniform approximation, [13] constructed a polynomial with degree $\widetilde{\Theta}(1/\eta^2)$ and further proved the existence of a $\widetilde{\Theta}(1/\eta)$-degree polynomial[4] but without algorithmic construction, which was done by [1]. Interestingly, a nearly matching lower bound on the degree had been shown by [12] much prior to these constructions.

# 3 Preliminaries

## 3.1 Notations

For a vector $x$, we use $\|x\|_p$ to denote its $\ell_p$ norm, and we are mostly concerned with $p = 1, 2$, or $\infty$ in this paper. For a matrix $W \in \mathbb{R}^{d \times m}$, we use $W^\top$ to denote the transpose of $W$, we use $\|W\|_F$, $\|W\|_1$ and $\|W\|$ to denote its Frobenius norm, entry-wise $\ell_1$ norm, and spectral norm respectively. We define $\|W\|_{2,\infty} = \max_{j \in [d]} \|W_j\|_2$, and $\|W\|_{2,1} = \sum_{j=1}^d \|W_j\|_2$, where $W_j$ is the $j$-th column of $W$, for each $j \in [m]$. We use $\mathcal{N}(\mu, \Sigma)$ to denote Gaussian distribution with mean $\mu$ and covariance $\Sigma$. We denote by $\sigma(\cdot)$ the ReLU function $\sigma(z) = \max\{z, 0\}$ and by $\mathbb{1}\{E\}$ the indicator function for an event $E$.

## 3.2 Two-layer ReLU network

We consider a two-layer ReLU activated neural network with $m$ neurons in the hidden layer:

$$f(x) = \sum_{r=1}^m a_r \sigma\left(\langle W_r, x \rangle + b_r\right) \tag{1}$$

where $W = (W_1, \ldots, W_m) \in \mathbb{R}^{d \times m}$ is the hidden weight matrix, $b = (b_1, \ldots, b_m) \in \mathbb{R}^m$ is the bias vector, and $a = (a_1, \ldots, a_m) \in \mathbb{R}^m$ is the output weight vector. We use $\mathcal{F}$ to denote this function class. During adversarial training, we only update $W$ and keep $a$ and $b$ at initialization values. For this reason, we write the network as $f_W(x)$.

We have $n$ training data $\mathcal{S} = \{(x_1, y_1), \ldots, (x_n, y_n)\} \subseteq \mathbb{R}^d \times \mathbb{R}$. We make some standard assumptions about the training set. Without loss of generality, we assume that for all $i \in [n]$, $\|x_i\|_2 = 1$ and the last coordinate $x_{i,d} = 1/2$ [5]. For this reason, we define the set $\mathcal{X} := \{x \in \mathbb{R}^d : \|x\|_2 = 1, \ x_d = 1/2\}$. We also assume for simplicity that for all $i \in [n]$, $|y_i| \leq 1$. The initialization of $a, W, b$ is $a^{(0)}, W^{(0)}, b^{(0)}$: the entries of $W^{(0)}$ and $b^{(0)}$ are iid random Gaussians from $\mathcal{N}(0, \frac{1}{m})$, and the entries of $a^{(0)}$ are iid with distribution $unif\left(\left\{-\frac{1}{m^{1/3}}, +\frac{1}{m^{1/3}}\right\}\right)$. [6]

### 3.3 Adversary and robust loss

To evaluate the neural nets, we consider a loss function of the following type.

**Definition 3.1** (Lipschitz convex regression loss). *A loss function $\ell : \mathbb{R} \times \mathbb{R} \to \mathbb{R}$ is a Lipschitz convex regression loss if it satisfies the following properties: convex in the first argument, non-negative, $1-$Lipshcitz and for all $y \in \mathbb{R}$, $\ell(y, y) = 0$.*

We remark the choice of loss is for simplicity of technical presentation, following the convention in previous works [14, 2].

For a vector $z \in \mathbb{R}^d$ and $\rho > 0$, let $\mathcal{B}_2(z, \rho) := \{x \in \mathbb{R}^d : \|x - z\|_2 \leq \rho\} \cap \mathcal{X}$. Note that in the paper we only consider $\ell_2$ perturbations for theoretical simplicity. Now we define the adversarial model studied in this paper.

**Definition 3.2** ($\rho$-Bounded adversary). *An adversary $\mathcal{A} : \mathcal{X} \times \mathbb{R} \times \mathcal{F} \to \mathcal{X}$ is $\rho$-bounded for $\rho > 0$ if they satisfy $\mathcal{A}(x, y, f) \in \mathcal{B}_2(x, \rho)$ We use $\mathcal{A}^*$ to denote the **worst-case** $\rho$-bounded adversary for loss function $\ell$, which is defined as $\mathcal{A}^*(x, y, f) := \arg\max_{\widetilde{x} \in \mathcal{B}_2(x, \rho)} \ell(f(\widetilde{x}), y)$ With a slight abuse of notation, we use $\mathcal{A}(S, f) := \{(\mathcal{A}(x_i, y_i, f), y_i)\}_{i=1}^n$ to denote the adversarial dataset generated by $\mathcal{A}$ against a given neural net $f$.*

We now define the robust loss of $f$ in terms of its prediction loss on the examples generated by an adversary.

**Definition 3.3** (Training loss and its robust version). *Given a training set $S$ of $n$ examples, the standard training loss of a neural net $f$ is defined as $\mathcal{L}(f, S) := \frac{1}{n} \sum_{i=1}^n \ell(f(x_i), y_i)$. Against a $\rho$-bounded adversary $\mathcal{A}$, we define the robust training loss w.r.t. $\mathcal{A}$ as $\mathcal{L}_{\mathcal{A}}(f) := \mathcal{L}(f, \mathcal{A}(S, f)) = \frac{1}{n} \sum_{i=1}^n \ell(f(\mathcal{A}(x_i, y_i, f)), y_i)$. Furthermore, we define analogously the **worst-case** robust training loss as $\mathcal{L}_{\mathcal{A}^*}(f) := \mathcal{L}(f, \mathcal{A}^*(S, f)) = \frac{1}{n} \sum_{i=1}^n \max_{\widetilde{x}_i \in \mathcal{B}_2(x_i, \rho)} \ell(f(\widetilde{x}_i), y_i)$*

### 3.4 Well-separated training sets

Training set being well-separated is a standard assumption in over-parametrization literature. Here we require a slightly stronger notion since we are dealing with adversarial perturbations.

**Definition 3.4** ($\gamma$-separability). *We say a training set $S$ is $\gamma$-separable with respect to a $\rho$-bounded adversary, if for all $i \neq j \in [n]$, $\|x_i - x_j\|_2 \geq \delta$ and $\gamma \leq \delta(\delta - 2\rho)$.*

Our results imply that the required width is polynomial for $\Omega(1)$-separable training sets. To see why this is a reasonable assumption, $\delta \approx \sqrt{3/2}$ if $x$'s are drawn from the uniform distribution on $\mathcal{X}$ and $d$ is large, while $\rho$ is usually at most $1/20$ in practice [15]. In Figure 1 we show that on CIFAR-10, other than probably a very small fraction of examples, all the others do not have too small minimum distance from any example.[7]

---

**Algorithm 1** Adversarial training

**Input:** Training set $\{(x_1, y_1), \ldots, (x_n, y_n)\}$, Adversary $\mathcal{A}$, learning rate $\eta$, initialization $a^{(0)}, W^{(0)}, b^{(0)}$.
    **for** $t = 0$ to $T - 1$ **do**
        $S^{(t)} := \emptyset$
        **for** $i = 1$ to $n$ **do**
            $\widetilde{x}_i^{(t)} = \mathcal{A}(x_i, y_i, f_{W^{(t)}})$
            $S^{(t)} = S^{(t)} \cup (\widetilde{x}_i^{(t)}, y_i)$
        **end for**
        $W^{(t+1)} = W^{(t)} - \eta \cdot \nabla_W \mathcal{L}(f_{W^{(t)}}, S^{(t)})$.
    **end for**
**Output:** $\{W^{(t)}\}_{t=1}^T$

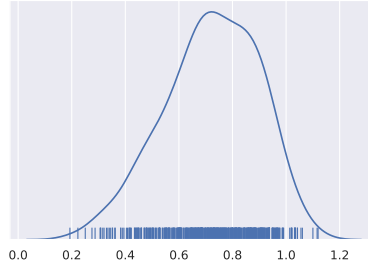

Figure 1: Distribution of $\delta_i$'s of randomly sampled 500 points in CIFAR-10 training set, where $\delta_i$ is the smallest $\ell_2$ distance between data point $x_i$ and any other point in the training set.

### 3.5 Adversarial training algorithm

The adversarial training of a neural net $f_W$ against an adversary $\mathcal{A}$ can be captured as the following intertwining dynamics. In the inner loop, the adversary generates adversarial examples against the current neural net. In the outer loop, a gradient descent step is taken on the neural net's parameter to decrease its prediction loss on the fresh adversarial examples.

**Remark.** *The gradient computation $\nabla_W \mathcal{L}(f_{W^{(t)}}, S^{(t)})$ is undertaken pretending as if $S^{(t)}$ was independent from $W^{(t)}$, i.e., without differentiating through $\mathcal{A}$.*

## 4 Main Result

We now formally present our main theorem.

**Theorem 4.1.** *Suppose that the training set $\mathcal{S}$ is $\gamma$-separable, for some $\gamma > 0$. Then, for all $\epsilon \in (0,1)$, there exist*

$$M_0 = \mathrm{poly}\left(d, \left(\frac{n}{\epsilon}\right)^{1/\gamma}\right) \ \text{and} \ R = \mathrm{poly}\left(\left(\frac{n}{\epsilon}\right)^{1/\gamma}\right)$$

*such that for every $m \geq M_0$, with probability at least $1 - \exp\left(-\Omega\left(m^{1/3}\right)\right)$ over the choice of $a^{(0)}, W^{(0)}, b^{(0)}$, if we run adversarial training 1 with hyper-parameters $T = \Theta(\epsilon^{-2}R^2)$ and $\eta = \Theta(\epsilon m^{-1/3})$, then the output weights $\left(W^{(t)}\right)_{t=1}^{T}$ satisfy $\min_{t \in [T]} \mathcal{L}_{\mathcal{A}}\left(f_{W^{(t)}}\right) \leq \epsilon$*

## 5 Proof Overview

**Pseudo-network** The key property used in all recent papers that analyze gradient descent for over-parameterized neural nets is that if a network $f_W(x) = \sum_{r=1}^{m} a_r^{(0)} \sigma\left(\langle W_r, x \rangle + b_r^{(0)}\right)$ is very over-parameterized and its weights are close to initialization, then it is well-approximated by its corresponding *pseudo-network*:

$$g_W(x) = \sum_{r=1}^{m} a_r^{(0)} \langle W_r - W_r^{(0)}, x \rangle \mathbb{1}\left\{\langle W_r^{(0)}, x \rangle + b_r^{(0)} \geq 0\right\}$$

However, the approximation result used for standard training is insufficient for our purposes, because here we deal with adversarial perturbations and in order to argue that during adversarial training the network behaves essentially as a pseudo-network, we need an approximation guarantee that holds *uniformly* over all $\mathcal{X}$. More specifically, in these works, it is proven that for any fixed input $x$, with probability at least $1 - e^{-\mathrm{poly}(\log m)}$, for $W$ close to the initialization, $|f_W(x) - g_W(x)|$ is small. But, with this probability bound, in order to argue that $\sup_{x \in \mathcal{X}} |f_W(x) - g_W(x)|$ is small via $\epsilon$-net arguments, one needs $m \geq \exp(\Omega(d))$. In this work, we show that the guarantee for fixed $x$ actually holds with much higher probability: $1 - \exp(-\Omega(m^{1/3}))$. The fact that this approximation fails with exponentially small probability, enables us to take a union bound over a very fine-grained $\frac{1}{\mathrm{poly}(m)}$-net of $\mathcal{X}$, and even though it has cardinality $\exp(O(d \log m))$, the width $m$ we need to control the overall probability is still polynomial in $d$. The final step is to bound the stability of $f$ and $g$ under small perturbations, even though $g$ is not Lipschitz continuous.

**Theorem 5.1.** *Let $R \geq 1$. For all $m \geq \mathrm{poly}(d)$, with probability at least $1 - \exp(-\Omega(m^{1/3}))$ over the choice of $a^{(0)}, W^{(0)}, b^{(0)}$, for all $W \in \mathbb{R}^{d \times m}$ such that $\|W - W^{(0)}\|_{2,\infty} \leq \frac{R}{m^{2/3}}$, $\sup_{x \in \mathcal{X}} |f_W(x) - g_W(x)| \leq O\left(\frac{R^2}{m^{1/6}}\right)$.*

We give the proof of Theorem 5.1 at the Appendix 10.1.

**Online convex optimization view** The adversarial training algorithm fits the framework of *online gradient descent* (OGD): at each step $t$, (1) the adversary chooses the loss function $\mathcal{L}_t(W) = \mathcal{L}\left(f_{W^{(t)}}, S^{(t)}\right)$, (2) the learner incurs the cost $\mathcal{L}_t(W^{(t)})$ and updates $W^{(t+1)} = W^{(t)} - \eta \nabla_W \mathcal{L}_t(W^{(t)})$. Online gradient descent comes with regret guarantees, when the loss functions are convex [16], but in our case they are not. However, it can be shown that during

adversarial training, the weights stay near initialization, which implies that the net behaves like a pseudo-net. Moreover, pseudo-net is linear in $W$ and so the regret guarantee holds, up to a small approximation error. Notably, the regret is with respect to the best net in hindsight, that is also close to initialization.

**Theorem 5.2.** *For all $\epsilon \in (0, 1)$, $R \geq 1$, there exists an $M = \text{poly}\left(n, R, \frac{1}{\epsilon}\right)$, such that for every $m \geq M$, with probability at least $1 - \exp\left(-\Omega\left(m^{1/3}\right)\right)$ over the choice of $a^{(0)}, W^{(0)}, b^{(0)}$, if we run Algorithm 1 with hyper-parameters $T = \Theta(\epsilon^{-2}R^2)$ and $\eta = \Theta(\epsilon m^{-1/3})$, then for every $W^*$ such that $\|W^* - W^{(0)}\|_{2,\infty} \leq \frac{R}{m^{2/3}}$, the output weights $\left(W^{(t)}\right)_{t=1}^T$ satisfy $\frac{1}{T}\sum_{t=1}^T \mathcal{L}_{\mathcal{A}}\left(f_{W^{(t)}}\right) \leq \mathcal{L}_{\mathcal{A}^*}\left(f_{W^*}\right) + \epsilon$*

Note that while in the LHS of the guarantee we have the robust losses w.r.t. $\mathcal{A}$, in the RHS we have the worst-case robust loss. We give the proof of Theorem 5.2 at the Appendix 10.2.

The connection with OCO was first made in [14]. However, they prove the above result for the case of quadratic ReLU activation. For the classical ReLU, they need to enforce the closeness to the initialization during training via a projection step, that is not used in practice.

**Existence of robust network near initialization**  What is left to do to prove Theorem 4.1 is to show the existence of a network $f_{W^*}$ that is close to initialization and the worst-case robust loss $\mathcal{L}_{\mathcal{A}^*}(f_{W^*})$ is small. [14] required $m$ to be at least $\left(\frac{1}{\epsilon}\right)^{\Omega(d)}$ to prove this statement. Our main result is the proof of existence of such network with width at most $\text{poly}\left(d, \left(\frac{n}{\epsilon}\right)^{1/\gamma}\right)$. Formally, for a $\rho$-bounded adversary and $\gamma$-separable training set, we have the following theorem.

**Theorem 5.3.** *For all $\epsilon \in (0, 1)$, there exists*

$$M_0 = \text{poly}\left(d, \left(\frac{n}{\epsilon}\right)^{1/\gamma}\right) \text{ and } R = \text{poly}\left(\left(\frac{n}{\epsilon}\right)^{1/\gamma}\right)$$

*such that for every $m \geq M_0$, with probability at least $1 - \exp\left(-\Omega\left(m^{1/3}\right)\right)$ over the choice of $a^{(0)}, W^{(0)}, b^{(0)}$, there exists $W^* \in \mathbb{R}^{d \times m}$ such that $\|W^* - W^{(0)}\|_{2,\infty} \leq \frac{R}{m^{2/3}}$ and $\mathcal{L}_{\mathcal{A}^*}(f_{W^*}) \leq \epsilon$.*

The proof of Theorem 5.3 has the following three steps, and we provide a sketch of the implementation of these three steps in section 6.

- We show that there is a function $f^* : \mathcal{X} \to \mathbb{R}$ that has "low complexity" and for all datapoints $(x_i, y_i)$ and perturbed inputs $\widetilde{x}_i \in \mathcal{B}_2(x_i, r)$, $f^*(\widetilde{x}_i) \approx y_i$. More specifically, this function will have the form $f^*(x) = \sum_{i=1}^n y_i q(\langle x_i, x \rangle)$ where $q$ is a low-degree polynomial approximating a step function that is 1 for $\langle x_i, x \rangle \approx 1$ and 0 otherwise. The existence of such a low-degree polynomial is proven using tools from approximation theory that appear in [23, 13].

- We show that since $f^*$ has "low complexity", there exists a pseudo-network $g_{W^*}$ that is close to initialization, has polynomial width (for $\gamma = \Omega(1)$), and $g_{W^*} \approx f^*$.

- We use Theorem 5.1 to show that for the real network $f_{W^*}$ we have $f_{W^*} \approx g_{W^*}$.

# 6   Proof of Theorem 5.3

We first provide the definition of a complexity measure for polynomials, following [2]. Note that the definitions of that paper also have an input parameter $R$. In this work, we set that $R$ to be 1.

**Definition 6.1.** *Let $c > 1$ denote a sufficiently large constant. For any degree-$k$ univariate polynomial $\phi(z) = \sum_{j=0}^k \alpha_j z^j$, and parameter $\epsilon_1 > 0$, we define the following two measures of complexity*

$$\mathfrak{C}(\phi, \epsilon_1) := \sum_{j=0}^k c^j \cdot (1 + (\sqrt{\ln(1/\epsilon_1)/j})^j) \cdot |\alpha_j|, \quad \mathfrak{C}(\phi) := c \cdot \sum_{j=0}^k (j+1)^{1.75}|\alpha_j|$$

## 6.1 Robust fitting with polynomials

In this section we show that the fact that the points $x_i$ in the training set have pairwise $\ell_2$ distance at least $\delta$ and $1/\delta$ is not too large implies that there is a function $f^*$ that has "low complexity" and robustly fits the training set: $\forall i \in [n], \widetilde{x}_i \in \mathcal{B}_2(x_i, \rho), \quad f^*(\widetilde{x}_i) \approx y_i$. Formally, we prove the following lemma.

**Lemma 6.2.** *Let $D = \frac{24}{\gamma} \ln \left( 48 \frac{n}{\epsilon} \right)$. There exists a polynomial $q : \mathbb{R} \to \mathbb{R}$ with degree at most $D$, size of coefficients at most $O(\gamma^{-1} 2^{6D})$, such that for all $j \in [n]$ and $\widetilde{x}_j \in \mathcal{B}_2(x_j, \rho)$, $|\sum_{i=1}^n y_i \cdot q(\langle x_i, \widetilde{x}_j \rangle) - y_i| \leq \frac{\epsilon}{3}$.*

Given the polynomial $q$ of the lemma, we will write $f^*(x) := \sum_{i=1}^n y_i \cdot q(\langle x_i, x \rangle)$. To prove Lemma 6.2, we first show how to approximate the step function via a polynomial. More specifically, the plan is this polynomial to take as input the inner product of two unit vectors $u, v$ and its output to be close to 1, if $\|u - v\|_2 \leq \rho$, and 0, if $\|u - v\|_2 \geq \delta - \rho$.

Note that since these are unit vectors, $\|u - v\|_2 \leq \rho$ is equivalent to $\langle u, v \rangle \geq 1 - \rho^2/2$, and $\|u - v\|_2 \geq \delta - \rho$ is equivalent to $\langle u, v \rangle \leq 1 - (\delta - \rho)^2/2$. We prove the following claim.

**Claim 6.3.** *Let $\epsilon_1 \in (0, 1)$ and $D = \frac{24}{\gamma} \ln \left( \frac{16}{\epsilon_1} \right)$. Then, there exists a univariate polynomial $q_{\epsilon_1}(z)$ with degree at most $D$ and size of coefficients at most $O(\gamma^{-1} 2^{6D})$, such that (1) $\forall z \in [1 - \rho^2/2, 1]$, $|q_{\epsilon_1}(z) - 1| \leq \epsilon_1$, and (2) $\forall z \in [-1, 1 - (\delta - \rho)^2/2)$, $|q_{\epsilon_1}(z)| \leq \epsilon_1$.*

*Proof.* For $\alpha \in [-1, 1]$, we define

$$\text{step}_\alpha(z) = \begin{cases} 0, & \text{if } -1 \leq z < \alpha \\ 1/2, & \text{if } z = \alpha \\ 1, & \text{if } \alpha < z \leq 1 \end{cases}, \quad \text{and} \quad \text{sgn}(z) = \begin{cases} -1, & \text{if } -1 \leq z < 0 \\ 0, & \text{if } z = 0 \\ 1, & \text{if } 0 < z \leq 1 \end{cases}$$

Note that $\text{step}_\alpha(z) = \frac{1}{2}(\text{sgn}(z - \alpha) + 1)$. We need a polynomial approximation result of the sgn function, from [13].

**Lemma 6.4** (Lemma 5.5 from [13]). *Let $\epsilon_1, \eta \in (0, 1)$ and $D = \frac{3}{\eta} \ln \frac{2}{\eta \epsilon_1}$. Then, there exists a univariate polynomial $p_{\epsilon_1}(z) = \sum_{j=0}^k \alpha_j z^j$ with degree $k \leq D$ and $|\alpha_j| \leq 2^{4D}$, that is an $\epsilon_1$-approximation of the sgn function in $[-1, 1] \setminus (-\eta, \eta)$, meaning that (1) $\forall z \in [\eta, 1], |p_{\epsilon_1}(z) - 1| \leq \epsilon_1$, and (2) $\forall z \in [-1, -\eta], |p_{\epsilon_1}(z) + 1| \leq \epsilon_1$.*

[13] describe how to construct the above polynomial and bound its degree, but do not present a bound on its coefficients. We prove Lemma 6.4 in Appendix 10.4. We can now approximate the step function by the polynomial

$$q_{\epsilon_1}(z) = \frac{p_{\epsilon_1}(2(z - \alpha)) + 1}{2}. \tag{2}$$

Because of the lemma and the connection between the sgn and the $\text{step}_\alpha$ functions, we get that $\forall z \in [-2 + \alpha, 2 + \alpha] \setminus [\alpha - 2\eta, \alpha + 2\eta], |q_{\epsilon_1}(z) - \text{step}_\alpha(z)| \leq \epsilon_1/2$.

Observe that $q_{\epsilon_1}$ also has degree $k$ and if $A = \max_j \{|\alpha_j|\}$, then the coefficient of $z^j$ in $q_{\epsilon_1}$ has size at most $2^{k-1} A \sum_{i=j}^k \binom{i}{j} |\alpha|^{i-j} + 1/2 \leq \frac{2^{2k-1} A}{1-\alpha} + 1/2 \leq \frac{2^{6D-1}}{1-\alpha} + 1/2$. Setting $\eta = \delta(\delta - 2\rho)/8 \leq \gamma/8$ and $\alpha = 1 - \frac{\rho^2}{2} - 2\eta$ finishes the proof. $\qquad \square$

To finish the proof of Lemma 6.2, let $q$ be the polynomial that we get from Claim 6.3, by setting $\epsilon_1 = \epsilon/(3n)$. Let $f^*(x) = \sum_{i=1}^n y_i q(\langle x_i, x \rangle)$. For all $i, j \in [n], i \neq j$ and $\widetilde{x}_i \in \mathcal{B}_2(x_i, \rho)$, we have $\|x_j - \widetilde{x}_i\|_2 \geq \delta - \rho$. Thus, from Claim 6.3 we have $|q(\langle x_j, \widetilde{x}_i \rangle)|, |q(\langle x_i, \widetilde{x}_i \rangle) - 1| \leq \epsilon/(3n)$.

$$|f^*(\widetilde{x}_i) - y_i| \leq |y_i||1 - q(\langle x_i, \widetilde{x}_i \rangle)| + \sum_{j \neq i} |y_j||q(\langle x_j, \widetilde{x}_i \rangle)| \leq \epsilon/(3n) + (n-1)\epsilon/(3n) \leq \epsilon/3$$

## 6.2 Pseudo-Network Approximates $f^*$

We prove that we can use a pseudo-network with width $\text{poly}\left(d, \left(\frac{n}{\epsilon}\right)^{1/\gamma}\right)$ to approximate $f^*$, uniformly over $\mathcal{X}$.

**Lemma 6.5.** *For all $\epsilon \in (0,1)$, there exist $M = \text{poly}\left(d, \left(\frac{n}{\epsilon}\right)^{1/\gamma}\right)$ and $R = \text{poly}\left(\left(\frac{n}{\epsilon}\right)^{1/\gamma}\right)$ such that for $m \geq M$, with probability at least $1 - \exp\left(-\Omega\left(\sqrt{m/n}\right)\right)$ over the choice of $a^{(0)}, W^{(0)}, b^{(0)}$, there exists there exists a $W^* \in \mathbb{R}^{d \times m}$ such that $\|W^* - W^{(0)}\|_{2,\infty} \leq \frac{R}{m^{2/3}}$ and $\sup_{x \in \mathcal{X}} |g_{W^*}(x) - f^*(x)| \leq \epsilon/3$.*

[2] prove a similar but weaker guarantee, by approximating $f^*$ using a pseudo-network, *in expectation*. In other words, they show that for some data distribution $\mathcal{D}$, $\mathbb{E}_{x \sim \mathcal{D}}\left[|g_{W^*}(x) - f^*(x)|\right]$ is small, for some pseudo-network $g_{W^*}$ close to initialization. As we mentioned previously, dealing with the average case is not enough and we need a uniform approximation guarantee, since we account for adversarial perturbations of the inputs.

We give here a proof sketch for Lemma 6.5 and the full proof at the Appendix 10.5. We use a technical result from [2]. Suppose that for a given unit vector $w^* \in \mathbb{R}^d$ and a univariate polynomial $\phi$, we want to approximate the function of a unit vector $x$ given by $\phi(\langle w^*, x \rangle)$, via a linear combination of random ReLU features. Intuitively, their result says that if $\phi$ has low complexity, then the weights of this linear combination can be small.

**Lemma 6.6** (Lemma 6.2 from [2]). *For every univariate polynomial $\phi : \mathbb{R} \to \mathbb{R}$, for every $\epsilon_2 \in (0, 1/\mathfrak{C}(\phi))$, there exists a function $h : \mathbb{R}^2 \to [-\mathfrak{C}(\phi, \epsilon_2), \mathfrak{C}(\phi, \epsilon_2)]$ such that for all $w^*, x \in \mathbb{R}^d$ with $\|w^*\|_2 = \|x\|_2 = 1$, we have $\left| \mathbb{E}_{u \sim \mathcal{N}(0, I_d), \beta \sim \mathcal{N}(0,1)} \left[ \mathbb{1}\{\langle u, x \rangle + \beta \geq 0\} h(\langle w^*, u \rangle, \beta) \right] - \phi(\langle w^*, x \rangle) \right| \leq \epsilon_2$.*

The above lemma implies $f^*$ can be approximated by an "infinite" pseudo-network. We use concentration bounds to argue that there exists a pseudo-network $g_{W^*}$ with width $\text{poly}\left(d, \left(\frac{n}{\epsilon}\right)^{1/\gamma}\right)$, such that for any fixed input $x \in \mathcal{X}$, with probability at least $1 - \exp(-\Omega(\sqrt{m/n}))$, $g_{W^*}(x) \approx f^*(x)$. We conclude the argument via a union bound over a $\frac{1}{\text{poly}(m)}$-net of $\mathcal{X}$ and a perturbation analysis for $g$, similarly to the proof of Theorem 5.1.

## 6.3 Putting it all together

We will use Lemmas 6.2, 6.5 and Theorem 5.1 to prove Theorem 5.3. From Lemma 6.2 we get $f^*$. From Lemma 6.5 we get the $M$, $R$ and $W^*$ and combining with Theorem 5.1, we have that as long as $m \geq \max\{\text{poly}(d), M\}$, with probability at least $p := 1 - \exp(-\Omega(\sqrt{m/n})) - \exp(-\Omega(m^{1/3}))$, there exists a $W^* \in \mathbb{R}^{d \times m}$ such that $\|W^* - W^{(0)}\|_{2,\infty} \leq \frac{R}{m^{2/3}}$ and for all $x \in \mathcal{X}, |g_{W^*}(x) - f^*(x)| \leq \epsilon/3$ and $|f_{W^*}(x) - g_{W^*}(x)| \leq O\left(\frac{R^2}{m^{1/6}}\right)$. Thus, for all $i \in [n], \tilde{x}_i \in \mathcal{B}(x_i, \rho)$,

$$\ell(f_{W^*}(\tilde{x}_i), y_i) \leq |f_{W^*}(\tilde{x}_i) - y_i| \leq |f^*(\tilde{x}_i) - y_i| + |g_{W^*}(\tilde{x}_i) - f^*(\tilde{x}_i)| + |f_{W^*}(\tilde{x}_i) - g_{W^*}(\tilde{x}_i)|$$
$$\leq \frac{2\epsilon}{3} + O\left(\frac{R^2}{m^{1/6}}\right) \leq \epsilon$$

since $m \geq \text{poly}\left(d, \left(\frac{n}{\epsilon}\right)^{1/\gamma}\right)$, for a large enough polynomial. Thus, we have that $L_{\mathcal{A}^*}(f^*) \leq \epsilon$. As for the bound on the probability of success $p$, since $m \geq n^3$ (for large enough polynomial in the lower bound for $m$), we get $p \geq 1 - \exp(-\Omega(-m^{1/3}))$.

# 7 Conclusion and discussion

We have shown that under a natural separability assumption on the training data, adversarial training on polynomially wide two-layer ReLU networks always converges in polynomial time to small robust training loss, significantly improving previous results. This may serve as an explanation for small

loss achieved by adversarial training in practice. Central in our proof is an explicit construction of a robust net near initialization, utilizing ideas from polynomial approximation.

As a future direction, it would be nice to improve the current exponential in $1/\gamma$ width requirement to polynomial. Ideally, the width requirement would fall back to $\mathrm{poly}(1/\gamma)$ as in standard (non-adversarial) training setting when the perturbation radius $\rho$ approaches zero, which is missing in our construction. We believe it may require a better understanding of the expressivity of over-parameterized nets. Furthermore, a natural next step is to extend our results to multi-layer ReLU networks.

## 8 Broader Impact

This does not present any foreseeable societal consequence.

## 9 Acknowledgements

The authors acknowledge fundings from ONR, NSF, Simons Foundation, DARPA/SRC, AWS, Schmidt Foundation and IAS.

## Footnotes

[2]This is the activation function $(ReLU(x))^2$.

[3]These bounds appear in Corollary $C.1$ in their paper.

[4]$\widetilde{\Theta}(\cdot)$ excludes logarithmic factors.

[5]$1/2$ can be padded to the last coordinate, $\|x_i\|_2 = 1$ can always be ensured from $\|x_i\|_2 \leq 1$ by padding $\sqrt{1 - \|x\|_2^2}$. Our analysis can still hold without padding by including bias terms in the hidden layer.

[6]In NTK literature, $a_r^0$ are distributed as $N(0, \epsilon^2)$ (or $unif(\{-\epsilon, +\epsilon\})$), for a small $\epsilon$. Here, we need $\epsilon = m^{-c}$, for some constant $c$ that can take a range of values. The $c$ that optimizes our bounds is $c = 1/3$.

[7] One can always exclude this small fraction from the training set and then suffer this fraction at the final robust 0-1 loss.

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
