[Supplementary Material]

## 10  Appendix

For functions $f, g : \mathcal{X} \to \mathbb{R}$, we define

$$\|f - g\|_\infty = \sup_{x \in \mathcal{X}} |f(x) - g(x)| \tag{3}$$

### 10.1  Proof of Theorem 5.1

*Proof.* Let $W \in \mathbb{R}^{d \times m}$, $\|W - W^{(0)}\|_{2,\infty} \leq \frac{R}{m^{2/3}}$, that is arbitrarily correlated with the initialization $a^{(0)}, W^{(0)}, b^{(0)}$. It suffices to bound $\|f_W - g_W\|_\infty$, where $\|\cdot\|_\infty$ is defined in 3. From now on we work with this $W$ and we write $f, g$ for $f_W$, $g_W$. Also, let $\Delta W_r = W_r - W_r^{(0)}$, $\mathbb{I}_{x,r}^{(0)} = \mathbb{1}\{\langle W_r^{(0)}, x\rangle + b_r^{(0)} \geq 0\}$ and $\mathbb{I}_{x,r} = \mathbb{1}\{\langle W_r^{(0)} + \Delta W_r, x\rangle + b_r^{(0)} \geq 0\}$. So, we write

$$f(x) = \sum_{r=1}^m a_r^{(0)} \left( \langle W_r^{(0)} + \Delta W_r, x\rangle + b_r^{(0)} \right) \mathbb{I}_{x,r}$$

$$g(x) = \sum_{r=1}^m a_r^{(0)} \langle \Delta W_r, x\rangle \mathbb{I}_{x,r}^{(0)}$$

We prove an elementary anti-concentration property of the Gaussian distribution.

**Claim 10.1.** *Let $u \sim \mathcal{N}(0, I_d)$ and $\beta \sim \mathcal{N}(0, 1)$, which are independent. For all $x \in \mathcal{X}$ and $t \geq 0$,*

$$\Pr[|\langle u, x\rangle + \beta| \leq t] = O(t).$$

*Proof.* We fix $x$ and $t$ and we have that $\langle u, x\rangle + \beta \sim \mathcal{N}(0, 2)$. Moreover,

$$\Pr_{z \sim \mathcal{N}(0,2)} \left[ |z| \leq t \right] = \int_{-t}^t \frac{1}{\sqrt{2\pi}} e^{-z^2/4} \mathrm{d}z \leq \sqrt{\frac{2}{\pi}} t$$

$\square$

For all $x \in \mathcal{X}$, $r \in [m]$ and $t \in \mathbb{R}_+$, we define

$$\Lambda_r(x, t) := \mathbb{1}\left\{ |\langle W_r^{(0)}, x\rangle + b_r^{(0)}| \leq t \right\}.$$

and observe that from Claim 10.1, after scaling by $\sqrt{m}$, we have that $\Pr[\Lambda_r(x, t) = 1] \leq O(t\sqrt{m})$. We will prove that for every fixed $x \in \mathcal{X}$, with high probability, $|f(x) - g(x)|$ is small.

**Lemma 10.2.** *For all $x \in \mathcal{X}$, with probability at least $1 - \exp(-\Omega(m^{1/3}))$,*

$$|f(x) - g(x)| \leq O(R^2/m^{1/6}) \tag{4}$$

*Proof.* Let $A_r := \mathbb{1}\{\mathbb{I}_{x,r} \neq \mathbb{I}_{x,r}^{(0)}\}$. We bound the size of $\sum_{r=1}^m A_r$ with the following claim.

**Claim 10.3.** *For all $x \in \mathcal{X}$, with probability at least $1 - \exp\left(-\Omega\left(m^{5/6}\right)\right)$,*

$$\sum_{r=1}^m A_r \leq O(R \cdot m^{5/6})$$

*Proof.* We fix an $x \in \mathcal{X}$. Since $\|x\|_2 = 1$ and $\|\Delta W\|_{2,\infty} \leq R/m^{2/3}$, we have that

$$A_r \leq \mathbb{1}\left\{ |\langle W_r^{(0)}, x\rangle + b_r^{(0)}| \leq \|\Delta W_r\|_2 \right\} \leq \Lambda_r(x, R/m^{2/3}).$$

But, as we mentioned previously, gaussian anti-concentration implies that

$$\Pr\left[\Lambda_r(x, R/m^{2/3}) = 1\right] \leq O(R/m^{1/6})$$

Since for our fixed $x$, these are $[m]$ independent Bernoulli random variables, standard concentration implies that with probability at least $1 - \exp(-\Omega(m^{5/6}))$,

$$\sum_{r=1}^{m} \Lambda_r(x, R/m^{2/3}) \leq O(Rm^{5/6}).$$

The fact that $\sum_{r=1}^{m} A_r \leq \sum_{r=1}^{m} \Lambda_r(x, R/m^{2/3})$ finishes the proof of the claim. $\qquad\square$

We decompose $f$, using the following three functions

**Definition 10.4.** *We define $f_1, f_2, f_3$ as follows:*

$$f_1(x) := \sum_{r=1}^{m} a_r^{(0)} \langle \Delta W_r, x \rangle \mathbb{I}_{x,r}$$

$$f_2(x) := \sum_{r=1}^{m} a_r^{(0)} (\langle W_r^{(0)}, x \rangle + b_r^{(0)}) \mathbb{I}_{x,r}^{(0)}$$

$$f_3(x) := \sum_{r=1}^{m} a_r^{(0)} (\langle W_r^{(0)}, x \rangle + b_r^{(0)})(\mathbb{I}_{x,r} - \mathbb{I}_{x,r}^{(0)})$$

It is easy to see that $f(x) = f_1(x) + f_2(x) + f_3(x)$. We proceed by showing that $|f_1(x) - g(x)|, |f_2(x)|$ and $|f_3(x)|$ are all small.

**Claim 10.5.** *With probability at least $1 - \exp(-\Omega(m^{5/6}))$,*

$$|f_1(x) - g(x)| \leq O(R^2/m^{1/6})$$

*Proof.* From the definition of $A_r$ we have that $|\mathbb{I}_{x,r} - \mathbb{I}_{x,r}^{(0)}| \leq A_r$.

$$\begin{aligned}
|f_1(x) - g(x)| &= \left| \sum_{r=1}^{m} a_r \langle \Delta W_r, x \rangle (\mathbb{I}_{x,r} - \mathbb{I}_{x,r}^{(0)}) \right| \\
&\leq \sum_{r=1}^{m} |a_r| \cdot |\langle \Delta W_r, x \rangle| \cdot A_r \\
&\leq \frac{R}{m} \sum_{r=1}^{m} A_r
\end{aligned}$$

The last step follows from $\|\Delta W\|_{2,\infty} \leq \frac{R}{m^{2/3}}$, $a_r \sim \{\pm \frac{1}{m^{1/3}}\}$. From Claim 10.3, with probability at least $1 - \exp(-\Omega(m^{5/6}))$,

$$|f_1(x) - g(x)| \leq O(R^2/m^{1/6})$$

$\qquad\square$

**Claim 10.6.** *With probability at least $1 - \exp(-\Omega(m^{1/3}))$,*

$$|f_2(x)| \leq O(1/m^{1/6})$$

*Proof.* From the definition of $f_2$,

$$f_2(x) = \sum_{r=1}^{m} a_r^{(0)} \sigma(\langle W_r^{(0)}, x \rangle + b_r^{(0)}).$$

By definition of the ReLU function $\sigma(\cdot)$,

$$\sum_{r=1}^{m} \sigma^2(\langle W_r^{(0)}, x \rangle + b_r^{(0)}) \leq \sum_{r=1}^{m} (\langle W_r^{(0)}, x \rangle + b_r^{(0)})^2$$

Also, note that for $r \in [m]$, $\langle W_r^{(0)}, x \rangle + b_r^{(0)} \sim \mathcal{N}(0, 2/m)$ and independent. From concentration of the sum of independent Chi-Square random variables, we have that with probability at least $1 - \exp(-\Omega(m))$,

$$\sum_{r=1}^{m} \sigma^2(\langle W_r^{(0)}, x \rangle + b_r^{(0)}) \leq \sum_{r=1}^{m} (\langle W_r^{(0)}, x \rangle + b_r^{(0)})^2 \tag{5}$$

$$= O(1) \tag{6}$$

Now, because of independence, using Hoeffding's concentration inequality, for some large constant $c > 0$,

$$\Pr\left[ \left| \sum_{r=1}^{m} a_r^{(0)} \sigma(\langle W_r^{(0)}, x \rangle + b_r^{(0)}) \right| \geq \frac{c}{m^{1/6}} \;\middle|\; W^{(0)}, b^{(0)} \right]$$

$$\leq \exp\left( -\Omega\left( \frac{m^{-1/3}}{\frac{1}{m^{2/3}} \sum_{r=1}^{m} \sigma^2\left( \langle W_r^{(0)}, x \rangle + b_r^{(0)} \right)} \right) \right)$$

and using the previous bound we get that overall, with probability at least $1 - \exp(-\Omega(m^{1/3}))$,

$$|f_2(x)| \leq O(1/m^{1/6})$$

$\square$

**Claim 10.7.** *With probability at least* $1 - \exp(-\Omega(m^{5/6}))$,

$$|f_3(x)| \leq O(R^2/m^{1/6})$$

*Proof.*

$$|f_3(x)| = \left| \sum_{r=1}^{m} a_r^{(0)} (\langle W_r^{(0)}, x \rangle + b_r^{(0)})(\mathbb{I}_{x,r} - \mathbb{I}_{x,r}^{(0)}) \right|$$

$$\leq \sum_{r=1}^{m} |a_r^{(0)}| \left| \langle W_r^{(0)}, x \rangle + b_r^{(0)} \right| \left| \mathbb{I}_{x,r} - \mathbb{I}_{x,r}^{(0)} \right|$$

$$\leq \frac{1}{m^{1/3}} \sum_{r=1}^{m} \left| \langle W_r^{(0)}, x \rangle + b_r^{(0)} \right| \left| \mathbb{I}_{x,r} - \mathbb{I}_{x,r}^{(0)} \right|$$

We use that

$$\left| \mathbb{I}_{x,r} - \mathbb{I}_{x,r}^{(0)} \right| \leq A_r \leq \Lambda_r(x, R/m^{2/3}).$$

Now, remember that $\Lambda_r(x, R/m^{2/3}) \neq 0 \iff \left| \langle W_r^{(0)}, x \rangle + b_r^{(0)} \right| \leq R/m^{2/3}$, so

$$\left| \langle W_r^{(0)}, x \rangle + b_r^{(0)} \right| \left| \mathbb{I}_{x,r} - \mathbb{I}_{x,r}^{(0)} \right| \leq \left| \langle W_r^{(0)}, x \rangle + b_r^{(0)} \right| \Lambda_r(x, R/m^{2/3}) \leq \frac{R}{m^{2/3}} \Lambda_r(x, R/m^{2/3})$$

Thus,

$$|f_3(x)| \leq \frac{R}{m} \sum_{r=1}^{m} \Lambda_r(x, R/m^{2/3}).$$

But, as we previously showed, with probability at least $1 - \exp(-\Omega(m^{5/6}))$,

$$\sum_{r=1}^{m} \Lambda_r(x, R/m^{2/3}) \leq O(Rm^{5/6}).$$

Thus, with probability at least $1 - \exp(-\Omega(m^{5/6}))$,

$$|f_3(x)| = O(R^2/m^{1/6}).$$

$\square$

We are ready to finish the proof of the lemma 4. Aggregating these three claims with a union bound, we have that for every $x \in \mathcal{X}$, with probability at least $1 - \exp(-\Omega(m^{5/6})) - \exp(-\Omega(m^{1/3})) = 1 - \exp(-\Omega(m^{1/3}))$, we have

$$|f(x) - g(x)| \leq |f_1(x) - g(x)| + |f_2(x)| + |f_3(x)| \leq O(R^2/m^{1/6}) \tag{7}$$

$\square$

What is left to do is to "union bound" over all $\mathcal{X}$. Of course, there is the problem that $\mathcal{X}$ is uncountable. So, we first do a union bound over a very fine-grained net of $\mathcal{X}$ and then argue about the change of $f$ and $g$ when we slightly change the input $x$.

Let $\mathcal{X}_1$ be a maximal $\frac{1}{m}$-net of $\mathcal{X}$. It is well-known that $|\mathcal{X}_1| \leq \left(\frac{1}{m}\right)^{O(d)}$. From lemma 4, by applying a union bound over $\mathcal{X}_1$, we have that for $m \geq cd^3$, where $c$ is a large constant, with probability at least

$$1 - \exp(O(d \log m)) \cdot \exp(-\Omega(m^{1/3})) = 1 - \exp(-\Omega(m^{1/3})),$$

we have

$$\forall x \in \mathcal{X}_1, \quad |f(x) - g(x)| \leq O(R^2/m^{1/6}) \tag{8}$$

The final step is the perturbation analysis. We show the following lemma, that applies for fixed inputs.

**Lemma 10.8.** *For all $x \in \mathcal{X}_1$, with probability at least $1 - \exp(-\Omega(m^{1/2}))$, for all $v \in \mathbb{R}^d$, such that $x + v \in \mathcal{X}$ and $\|v\|_2 \leq \frac{1}{m}$, we have*

$$|f(x + v) - f(x)| \leq O(1/m^{1/3} + R/m) \tag{9}$$

*and*

$$|g(x + v) - g(x)| \leq O(R/m^{1/2}). \tag{10}$$

With this lemma at hand, we can do a union bound over $\mathcal{X}_1$ and conclude that with probability at least $1 - \exp(O(d \log m)) \exp\left(-\Omega\left(m^{1/2}\right)\right) = 1 - \exp\left(-\Omega\left(m^{1/2}\right)\right)$ (since $m \geq cd^3$ and $c$ is a large constant), we have that for all $x \in \mathcal{X}_1$ and $v \in \mathbb{R}^d$, such that $x + v \in \mathcal{X}$ and $\|v\|_2 \leq \frac{1}{m}$, the perturbation guarantees 9 and 10 hold. Combining this with 8 and applying a union bound, we have that with probability at least $1 - \exp(-\Omega(m^{1/3})) - \exp\left(-\Omega\left(m^{1/2}\right)\right) = 1 - \exp(-\Omega(m^{1/3}))$,

$$\|f - g\|_\infty \leq O(R^2/m^{1/6} + 1/m^{1/3} + R/m + R/m^{1/2}) = O(R^2/m^{1/6})$$

and this concludes the proof of theorem 5.1.

$\square$

It remains to prove the Lemma 10.8.

Let $v$ be a small perturbation of $x$ with the properties stated in the lemma, that can depend arbitrarily on $a^{(0)}, W^{(0)}, b^{(0)}$.

$$|f(x+v) - f(x)| = \left| \sum_{r=1}^{m} a_r^{(0)} \left( \sigma \left( \langle W_r^{(0)} + \Delta W_r, x+v \rangle + b_r^{(0)} \right) - \sigma \left( \langle W_r^{(0)} + \Delta W_r, x \rangle + b_r^{(0)} \right) \right) \right|$$

$$\leq \sum_{r=1}^{m} |a_r^{(0)}| \left| \langle W_r^{(0)} + \Delta W_r, v \rangle \right|$$

$$\leq \frac{1}{m} \sum_{r=1}^{m} |a_r^{(0)}| \| W_r^{(0)} + \Delta W_r \|_2$$

$$= \frac{1}{m^{1+1/3}} \sum_{r=1}^{m} \| W_r^{(0)} + \Delta W_r \|_2$$

$$\leq \frac{1}{m^{4/3}} \sum_{r=1}^{m} \| W_r^{(0)} \|_2 + \frac{1}{m^{4/3}} \sum_{r=1}^{m} \| \Delta W_r \|_2$$

$$\leq \frac{1}{m^{4/3}} \sum_{r=1}^{m} \| W_r^{(0)} \|_2 + \frac{R}{m}$$

We show the following claim, which concludes the proof of 9.

**Claim 10.9.** *With probability at least $1 - \exp(-\Omega(m))$, $\| W^{(0)} \|_{2,\infty} \leq O(1)$.*

*Proof.* From concentration of sum of independent Chi-Square random variables, we have that for all $r$, with probability at least $1 - \exp(-\Omega(m^2/d))$, $\| W_r^{(0)} \|_2^2 \leq O(1)$. Since $m \geq d$, a union bound over all $r$ finishes the proof of the claim. $\qquad\square$

We now argue about g.

$$|g(x+v) - g(x)|$$

$$= \left| \sum_{r=1}^{m} a_r^{(0)} \langle \Delta W_r, x+v \rangle \mathbb{1}\{ \langle W_r^{(0)}, x+v \rangle + b_r^{(0)} \geq 0 \} - \sum_{r=1}^{m} a_r^{(0)} \langle \Delta W_r, x \rangle \mathbb{1}\{ \langle W_r^{(0)}, x \rangle + b_r^{(0)} \geq 0 \} \right|$$

$$\leq \frac{1}{m} \sum_{r=1}^{m} |a_r^{(0)}| \| \Delta W_r \|_2 + \sum_{r=1}^{m} |a_r^{(0)}| \, |\langle \Delta W_r, x \rangle| \, \left| \mathbb{1}\{ \langle W_r^{(0)}, x+v \rangle + b_r^{(0)} \geq 0 \} - \mathbb{1}\{ \langle W_r^{(0)}, x \rangle + b_r^{(0)} \geq 0 \} \right|$$

$$\leq \frac{R}{m} + \frac{R}{m} \sum_{r=1}^{m} \left| \mathbb{1}\{ \langle W_r^{(0)}, x+v \rangle + b_r^{(0)} \geq 0 \} - \mathbb{1}\{ \langle W_r^{(0)}, x \rangle + b_r^{(0)} \geq 0 \} \right|.$$

About the last sum, from Claim 10.9, $\| W^{(0)} \|_{2,\infty} \leq O(1)$ and in this case,

$$\sum_{r=1}^{m} \left| \mathbb{1}\left\{ \langle W_r^{(0)}, x+v \rangle + b_r^{(0)} \geq 0 \right\} - \mathbb{1}\{ \langle W_r^{(0)}, x \rangle + b_r^{(0)} \geq 0 \} \right| \leq \sum_{r=1}^{m} \Lambda_r \left( x, O(1/m) \right)$$

From Claim 10.1, we have that $\Lambda_r(x, O(1/m)) = 1$ with probability at most $O\left( \frac{1}{m^{1/2}} \right)$. Since $x$ is fixed, these are $m$ independent Bernoulli random variables and from standard concentration, with probability at least $1 - \exp(-\Omega(\sqrt{m}))$,

$$\sum_{r=1}^{m} \Lambda_r(x, O(1/m)) \leq O(\sqrt{m}).$$

This finishes the proof of 10.

## 10.2 Proof of Theorem 5.2

*Proof.* We will give the values of $T$ and $\eta$, later in the proof. For simplicity, we use the following shorthand notations to denote various distances.

$$D_{\max} := \max_{t \in [T]} \|W^{(t)} - W^{(0)}\|_{2,\infty}$$

$$D_{W^*} := \|W^* - W^{(0)}\|_{2,\infty}$$

By condition, we know $D_{W^*} = O\left(\frac{R}{m^{2/3}}\right)$.

Even though in Algorithm 1 the parameters $W$ are updated using the gradients of the *real net*, in this proof we consider the pseudo-net as the object being optimized. Thus we need to relate the real net gradients to the pseudo-net gradients. For ease of presentation, we define the following convenient notations for the two notions of gradients:

$$\text{real net gradient } \nabla^{(t)} := \nabla_W \mathcal{L}(f(W^{(t)}), S^{(t)})$$

$$\text{pseudo-net gradient } \widehat{\nabla}^{(t)} := \nabla_W \mathcal{L}(g(W^{(t)}), S^{(t)})$$

We write both gradients as matrices in $\mathbb{R}^{d \times m}$ In fact, by Lemma 10.10, we know that they are coupled with high probability, as long as $W^{(t)}$ stays close to initialization (i.e., $D_{\max} \leq m^{-15/24}$).

$$\|\widehat{\nabla}^{(t)} - \nabla^{(t)}\|_{2,1} \leq O\left(nm^{13/24}\right)$$

**Remark.** *We assume for now $D_{\max} \leq m^{-15/24}$ is true and in the end we will set proper values for $T, \eta$ and $m$ to make sure this is indeed the case.*

Using the fact that the loss is 1-Lipschitz, we bound the gradient size:

$$\|\nabla_r^{(t)}\|_2 \leq |a_r| \left(\frac{1}{n} \sum_{i=1}^{n} \sigma'\left(\langle W_r^{(t)}, x_i \rangle + b_r^{(0)}\right) \|\widetilde{x}_i\|_2\right) \leq \frac{1}{m^{1/3}} \tag{11}$$

Due to the linearity of $g$ with respect to $W$, the loss $\mathcal{L}(g(W), S)$ is convex in $W$. For two matrices $A, B$ with the same dimensions, we write their inner product as $\langle A, B, : \rangle = \text{tr}(A^T B)$.

$$\mathcal{L}(g(W^{(t)}), S^{(t)}) - \mathcal{L}(g(W^*), S^{(t)})$$
$$\leq \langle \nabla^{(t)}, W^{(t)} - W^* \rangle + \langle \widehat{\nabla}^{(t)} - \nabla^{(t)}, W^{(t)} - W^* \rangle$$
$$\leq \underbrace{\langle \nabla^{(t)}, W^{(t)} - W^* \rangle}_{:=\alpha^{(t)}} + \underbrace{\|\widehat{\nabla}^{(t)} - \nabla^{(t)}\|_{2,1} \|W^{(t)} - W^*\|_{2,\infty}}_{:=\beta^{(t)}}$$

We deal with $a^{(t)}$ and $b^{(t)}$ terms separately. As for the former, we use the standard online gradient descent proof technique:

$$\|W^{(t+1)} - W^*\|_F^2 = \|W^{(t)} - \eta \nabla^{(t)} - W^*\|_F^2 = \|W^{(t)} - W^*\|_F^2 - 2\eta \alpha^{(t)} + \eta^2 \|\nabla^{(t)}\|_F^2$$

So, by rearranging we get

$$\alpha^{(t)} \leq \frac{\eta}{2} \|\nabla^{(t)}\|_F^2 + \frac{\|W^{(t)} - W^*\|_F^2 - \|W^{(t+1)} - W^*\|_F^2}{2\eta}$$

and then sum over $t$,

$$\sum_{t=1}^{T} \alpha^{(t)} \leq \frac{\eta}{2} \sum_{t=1}^{T} \|\nabla^{(t)}\|_F^2 + \frac{\|W^{(0)} - W^*\|_F^2 - \|W^{(T+1)} - W^*\|_F^2}{2\eta} \leq \frac{\eta m^{1/3}}{2} T + \frac{m D_{W^*}^2}{2\eta}$$

where we used the fact $\|W^* - W^{(0)}\|_F^2 \leq m \cdot \|W^* - W^{(0)}\|_{2,\infty} = m D_{W^*}^2$ as well as $\|\nabla^{(t)}\|_F^2 \leq \sum_{r=1}^{m} \|\nabla_r^{(t)}\|_2^2 \leq m^{1/3}$.

For the $\beta^{(t)}$'s, we first invoke Lemma 10.10 and then apply triangle inequality:

$$\beta^{(t)} \leq O\left(nm^{13/24}\right) \|W^{(t)} - W^*\|_{2,\infty} \leq O\left(nm^{13/24}\right)(D_{\max} + D_{W^*})$$

Furthermore we can bound the size of $D_{\max}$ using the bound on gradients, i.e. $\|\nabla_r^{(t)}\|_2 \leq m^{-1/3}$ using inequality 11.

$$D_{\max} = \max_{t \in [T]} \|W^{(0)} - W^{(t)}\|_{2,\infty} \leq \sum_{t=1}^{T} \eta \max_{r \in [m]} \|\nabla_r^{(t)}\|_2 \leq \frac{\eta T}{m^{1/3}}$$

Putting it together with the condition $D_{W^*} = O\left(\frac{R}{m^{2/3}}\right)$ that we already have, we obtain the following:

$$\sum_{t=1}^{T} \mathcal{L}(g(W^{(t)}), S^{(t)}) - \sum_{t=1}^{T} \mathcal{L}(g(W^*), S^{(t)})$$

$$\leq \sum_{t=1}^{T} \alpha^{(t)} + \sum_{t=1}^{T} \beta^{(t)}$$

$$\leq O(1)\left(m^{1/3}\eta T + \frac{R^2}{m^{1/3}\eta} + \eta T n m^{5/24} + \frac{\eta R T n}{m^{1/8}}\right)$$

We then have

$$\frac{1}{T}\sum_{t=1}^{T} \mathcal{L}(g(W^{(t)}), S^{(t)}) - \frac{1}{T}\sum_{t=1}^{T} \mathcal{L}(g(W^*), S^{(t)}) \leq O(\epsilon)$$

if we set the hyper-parameters $T, m, \eta$ to be the following:

$$T = \Theta(\epsilon^{-2}R^2),$$

$$m \geq \Omega\left(\max\left\{n^8, \left(\tfrac{Rn}{\epsilon}\right)^{24/11}, \left(\tfrac{R^2}{\epsilon}\right)^{24}\right\}\right),$$

$$\eta = \tfrac{R}{m^{1/3}\sqrt{T}} = \Theta(m^{-1/3}\epsilon)$$

Note the the requirement on $m$ is to satisfy $\eta T n m^{1/4} + \frac{\eta R T n}{m^{1/12}} \leq O(\epsilon)$, $D_{\max} \leq m^{-15/24}$ as well as to meet the condition for invoking Theorem 5.1:

$$\forall t \in [T], \sup_{x \in \mathcal{X}} |f_{W^{(t)}}(x) - g_{W^{(t)}}(x)| \leq O(\epsilon)$$

Thus, we get

$$\frac{1}{T}\sum_{t=1}^{T} \mathcal{L}(f_{W^{(t)}}, S^{(t)}) - \frac{1}{T}\sum_{t=1}^{T} \mathcal{L}(f_{W^*}, S^{(t)}) \leq c \cdot \epsilon$$

where $c > 0$ is a large constant. Now, observe that $\mathcal{L}(f_{W^{(t)}}, S^{(t)}) = \mathcal{L}_{\mathcal{A}}(f_{W^{(t)}})$ and $\mathcal{L}(f_{W^*}, S^{(t)}) \leq \mathcal{L}_{\mathcal{A}^*}(f_{W^*})$. The proof we presented holds for all $\epsilon > 0$, so by using $\frac{\epsilon}{c}$ in place of $\epsilon$, we get the desired result. $\qquad\square$

## 10.3 Gradient coupling

**Lemma 10.10.** *With probability at least* $1 - \exp(-\Omega(m^{1/3}))$, *for all iterations t that* $\|W^{(t)} - W^{(0)}\|_{2,\infty} \leq O\left(m^{-15/24}\right)$, *we have*

$$\|\widehat{\nabla}^{(t)} - \nabla^{(t)}\|_{2,1} \leq O\left(nm^{13/24}\right)$$

*Proof.* We first prove the following claim.

**Claim 10.11.** *With probability probability at least* $1 - \exp(-\Omega(m^{1/3}))$ *over the initialization, for all subsets* $\{x_1, \ldots, x_n\} \subseteq \mathcal{X}$ *with $n$ points and any* $\|\Delta W_r\|_2 \leq m^{-15/24}$,

$$\sum_{r=1}^{m} \mathbb{1}\left\{\exists i \in [n], \text{ sgn}\left(\langle W_r^{(0)} + \Delta W_r, x_i\rangle + b_r^{(0)}\right) \neq \text{sgn}\left(\langle W_r^{(0)}, x_i\rangle + b_r^{(0)} \geq 0\right)\right\} \leq O\left(nm^{7/8}\right)$$

*Proof.* We first prove the above result for a fixed set of $n$ points, and then apply a union bound over all possible such sets. For a fixed set of $n$ points $\{x_1, \ldots, x_n\} \subseteq \mathcal{X}$, we define

$$B_r := \mathbb{1}\left\{\exists i \in [n], \text{ sgn}\left(\langle W_r^{(t)}, x_i\rangle + b_r^{(0)}\right) \neq \text{sgn}\left(\langle W_r^{(0)}, x_i\rangle + b_r^{(0)} \geq 0\right)\right\}$$

and the goal is to bound the size of $\sum_{r=1}^{m} B_r$.

We know by Claim 10.1 that for each $x_i$ we have

$$\Pr\left[|\langle W_r^0, x_i\rangle + b_r^{(0)}| \leq m^{-15/24}\right] \leq O\left(m^{-1/8}\right)$$

With a union bound over the indices $i \in [n]$, we have

$$\Pr\left[\exists i \in [n], \; |\langle W_r^0, x_i\rangle + b_r^{(0)}| \leq m^{-15/24}\right] \leq O\left(nm^{-1/8}\right)$$

which implies

$$\Pr[B_r = 1] \leq \Pr\left[\exists i \in [n], \; |\langle W_r^0, x_i\rangle + b_r^{(0)}| \leq m^{-15/24}\right] \leq O\left(nm^{-1/8}\right)$$

Because $x_i$'s are fixed for now, $B_r$'s are $m$ independent Bernoulli random variables. Standard concentration implies that with probability at least $1 - \exp(-\Omega(nm^{7/8}))$

$$\sum_{r=1}^{m} B_r \leq O\left(nm^{7/8}\right)$$

As a last step, we take a union bound over a $\frac{1}{m}$-net over product space $\otimes^n \mathcal{X}$ which amplifies the failure probability negligibly by only $\exp(O(nd \log m))$ compared to $\exp(-\Omega(m^{1/3}))$ (for large enough $m$). □

Now, we are ready to finish the proof of the coupling lemma. Remember that $D_{\max} = \|W^{(t)} - W^{(0)}\|_{2,\infty}$. By Claim 10.11, with probability at least $1 - \exp(-\Omega(m^{1/3}))$, all $t$,

$$\sum_{r=1}^{m} \mathbb{1}\left\{\nabla_r^{(t)} = \widehat{\nabla}_r^{(t)}\right\} \leq O\left(nm^{7/8}\right)$$

For the indices $r$'s that $\nabla_r^{(t)} \neq \widehat{\nabla}_r^{(t)}$, we have

$$\|\widehat{\nabla}_r^{(t)} - \nabla_r^{(t)}\|_2 \leq |a_r|\frac{1}{n}\sum_{i=1}^{n}\left|\mathbb{1}\{\langle W_r^{(t)}, \widetilde{x}_i\rangle + b_r^{(0)} \geq 0\} - \mathbb{1}\{\langle W_r^{(0)}, x_i\rangle + b_r^{(0)} \geq 0\}\right|\|\widetilde{x}_i\|_2$$

$$\leq \frac{1}{m^{1/3}}\frac{1}{n}\sum_{i=1}^{n}\left|\mathbb{1}\{\langle W_r^{(t)}, \widetilde{x}_i\rangle + b_r^{(0)} \geq 0\} - \mathbb{1}\{\langle W_r^{(0)}, x_i\rangle + b_r^{(0)} \geq 0\}\right|$$

$$\leq \frac{1}{m^{1/3}}$$

Thus, we conclude

$$\|\widehat{\nabla}^{(t)} - \nabla^{(t)}\|_{2,1} = \sum_{r=1}^{m}\|\widehat{\nabla}_r^{(t)} - \nabla_r^{(t)}\|_2 \leq \frac{1}{m^{1/3}} \cdot O\left(nm^{7/8}\right) = O\left(nm^{13/24}\right)$$

□

## 10.4 Proof of lemma 6.4

Let

$$p_k(z) := z \sum_{i=0}^{k} (1-z^2)^i \prod_{j=1}^{i} \frac{2j-1}{2j} \tag{12}$$

**Lemma 10.12** (Corollary 5.4 in [13])**.** *If $z \in [-1,1]$ with $|z| \geq \eta > 0$ and $k = \frac{1}{\eta^2} \ln(2/\epsilon_1)$, then $|\operatorname{sgn}(z) - p_k(z)| \leq \epsilon_1/2$. Moreover, $p_k$ has degree $2k+1$.*

We will now compress $p_k$ using Chebyshev polynomials. Recall that the Chebyshev polynomials of the first kind are defined as $T_0(z) = 1, T_1(z) = z$ and

$$T_{k+1}(z) = zT_k(z) - T_{k-1}(z) \tag{13}$$

The definition is also extended for negative $k$ as $T_{-k}(z) = T_k(z)$.

We will use the closed-form formula of $T_k(z)$:

$$T_k(z) = \sum_{i=0}^{\lfloor n/2 \rfloor} \binom{n}{2i} (z^2 - 1)^i z^{k-2i} \tag{14}$$

We bound the magnitude of the coefficients of the Chebyshev polynomials via the following proposition.

**Proposition 10.13.** *The magnitude of the coefficients of $T_k(z)$ is at most $2^{2k}$.*

*Proof.* From the closed-form formula in 14, we have that

$$T_k(z) = \sum_{i=0}^{\lfloor k/2 \rfloor} \binom{k}{2i} \sum_{j=0}^{i} \binom{i}{j} z^{2j} (-1)^{i-j} z^{k-2i} = \sum_{i=0}^{\lfloor k/2 \rfloor} \sum_{j=0}^{i} \binom{k}{2i}\binom{i}{j} (-1)^{i-j} z^{k+2j-2i}$$

The monomials that appear in the above polynomial are the $z^{k-2u}$, for $u = 0, \ldots, \lfloor k/2 \rfloor$. The magnitude of the coefficient of $z^{k-2u}$ is at most

$$\left| \sum_{i=u}^{\lfloor k/2 \rfloor} \binom{k}{2i}\binom{i}{i-u}(-1)^u \right| \leq \sum_{i=u}^{\lfloor k/2 \rfloor} \binom{k}{2i}\binom{i}{u} \leq \sum_{i=0}^{k} \binom{k}{i}\binom{k}{\lfloor k/2 \rfloor} \leq 2^{2k}$$

$\square$

Now, let $s$ be a positive integer, $Y_1, \ldots, Y_s$ iid $\pm 1$ random variables and $D_s := \sum_{i=1}^{s} Y_i$. Also, let $D \geq 0$. We define

$$p_{s,D}(z) := \mathbb{E}_{Y_1,\ldots,Y_s}[T_{D_s}(z) \mathbb{1}\{|D_s| \leq D\}] \tag{15}$$

A straightforward consequence of the proposition 10.13 is the following corollary.

**Corollary 10.14.** *$p_{s,D}(z)$ has degree at most $D$ and its coefficients have magnitude at most $2^{2D}$.*

We will use the following theorem from [23].

**Theorem 10.15** (Theorem 3.3 from [23])**.** *For all positive integers $s$, $D$ and for all $z \in [-1,1]$,*

$$|p_{s,D}(z) - z^s| \leq 2e^{-D^2/(2s)} \tag{16}$$

Now, we are ready to compress $p_k$. Let $\widetilde{p}_k(z) := \sum_{i=0}^{k} z \left( \prod_{j=1}^{i} \frac{2j-1}{2j} \right) p_{i,D}(1 - z^2)$. Also, let $D = \sqrt{2k \ln(4k/\epsilon_1)}$.

From the above theorem, we have that for all $z \in [-1, 1]$,

$$|\widetilde{p}_k(z) - p_k(z)| = \left| \sum_{i=0}^{k} z \left( \prod_{j=1}^{i} \frac{2j-1}{2j} \right) \left( p_{i,D}(1 - z^2) - (1 - z^2)^i \right) \right| \leq \sum_{i=0}^{k} \left| p_{i,D}(1 - z^2) - (1 - z^2)^i \right| \tag{17}$$

$$\leq \sum_{i=0}^{k} 2e^{-D^2/(2i)} \leq \epsilon_1/2 \tag{18}$$

Combining with lemma 10.12, we get that for $k = \frac{1}{\eta^2} \ln(2/\epsilon_1)$, for all $z \in [-1, 1]$, $|\operatorname{sgn}(z) - \widetilde{p}_k(z)| \leq \epsilon_1$. Let $p_\epsilon(z) := \widetilde{p}_k(z)$. We already know that the degree of $p_\epsilon(z)$ is at most $D = \frac{1}{\eta} \sqrt{2 \ln(2/\epsilon_1) \ln \left( 4 \frac{\ln(2/\epsilon_1)}{\eta^2 \epsilon_1} \right)} \leq \frac{3}{\eta} \ln(2/(\eta \epsilon_1))$.

It remains to bound the magnitude of its coefficients. Let $p_{i,D}(z) = \sum_{j=0}^{D} \alpha_j z^j$. From Corollary 10.14, we have that $\alpha_{max} := \max_j |\alpha_j| \leq 2^{2D}$. Now,

$$p_{i,D}(1 - z^2) = \sum_{j=0}^{D} \alpha_j (1 - z^2)^j = \sum_{j=0}^{D} \alpha_j \sum_{u=0}^{j} \binom{j}{u} (-1)^u z^{2u}$$

The magnitude of the coefficient of $z^{2u}$ is at most $(D + 1) \cdot \alpha_{max} \cdot \binom{D}{u} \leq (D + 1)2^{3D} \leq 2^{4D}$, since $D \geq 1.4$.

## 10.5 Proof of Lemma 6.5

We will first prove that we can approximate the individual components of $f^*$ via pseudo-networks and then we aggregate these to form a large pseudo-network that approximates $f^*$.

**Lemma 10.16.** *Let $i \in [n]$, $q : \mathbb{R} \to \mathbb{R}$ univariate polynomial and $\epsilon_3 \in \left( 0, \frac{1}{\mathfrak{C}(q)} \right)$. Let $\widetilde{m} \geq c_1 \frac{d}{\epsilon_3^2} \mathfrak{C}^2(q, \epsilon_3)$, for a large constant $c_1$. For all $r \in [\widetilde{m}]$, $U_r^{(0)} \sim \mathcal{N}(0, I_d)$, $\beta_r^{(0)} \sim \mathcal{N}(0, 1)$, $\alpha_r^{(0)} \sim unif\{\pm \frac{1}{m^{1/3}}\}$ and all these random variables and vectors are independent. With probability at least $1 - \exp\left( -\Omega\left( \sqrt{\widetilde{m}} \right) \right)$, there exists a matrix $\Delta W^{(i)} \in \mathbb{R}^{d \times \widetilde{m}}$ with $\|\Delta W^{(i)}\|_{2,\infty} \leq O\left( m^{1/3} \frac{\mathfrak{C}(q,\epsilon_3)}{\widetilde{m}} \right)$ such that*

$$\forall x \in \mathcal{X},$$
$$\left| \sum_{r=1}^{\widetilde{m}} \alpha_r^{(0)} \langle \Delta W_r^{(i)}, x \rangle \mathbb{1}\{\langle U_r^{(0)}, x \rangle + \beta_r^{(0)} \geq 0\} - y_i q(\langle x_i, x \rangle) \right| \leq 3\epsilon_3$$

With this Lemma at hand, we can finish the proof of Lemma 6.5. We apply it for all $i \in [n]$, with $q(z)$ being the polynomial that is given to us by Lemma 6.2. We now that the degree of $q$ is at most $D$ and the size of its coefficients is at most $c_2 \frac{1}{\gamma} 2^{6D}$ where $D = \frac{24}{\gamma} \ln(48n/\epsilon)$ and $c_2 > 0$ is a constant. Using this information about $q$, we can bound its complexities $\mathfrak{C}(q)$ and $\mathfrak{C}(q, \epsilon_3)$, defined in 6.1, where $\epsilon_3$ will be set after we bound $\mathfrak{C}(q)$ (since from Lemma 10.16 $\epsilon_3 < 1/\mathfrak{C}(q)$). About $\mathfrak{C}(q)$, we directly have $\mathfrak{C}(q) \leq c \cdot c_2 \sum_{j=0}^{D} (j + 1)^{1.75} \frac{1}{\gamma} 2^{6D} < c \cdot c_2 \frac{(D+1)^{2.75}}{\gamma} 2^{6D}$. We set $\epsilon_3 = \left( c \cdot c_2 \frac{(D+1)^{2.75}}{\gamma} 2^{6D} \right)^{-1}$. About $\mathfrak{C}(q, \epsilon_3)$, we have

$$\mathfrak{C}(q, \epsilon_3) \leq c_2 \sum_{j=0}^{D} c^j \left(1 + \sqrt{\ln(1/\epsilon_3)/j}\right)^j \frac{1}{\gamma} 2^{4D}$$

$$\leq O(1)\frac{1}{\gamma}2^{4D}(D+1)c^D e^{\sqrt{D \ln 1/\epsilon_3}}$$

$$= O(1)\frac{1}{\gamma}2^{4D}(D+1)c^D e^{\sqrt{D \ln\left(c \cdot c_2 \frac{(D+1)^{2.75}}{\gamma}2^{4D}\right)}}$$

$$\leq 2^{O(D)} \tag{19}$$

We specify now how we are performing the $n$ applications of the lemma, in terms of the choice of $\widetilde{m}$ and the random variables. Let $\widetilde{B} := \lceil c_1 \frac{d}{\epsilon_3^2} \mathfrak{C}^2(q, \epsilon_3) \rceil$. We use the fact that for large enough constant $c$, $m \geq d\left(\frac{n}{\epsilon}\right)^{c/\gamma} \geq n\widetilde{B}$. For $i = 1, \cdots, n-1$ we apply the lemma 10.16 with $\widetilde{m} = \lfloor \frac{m}{n} \rfloor$ and for $i = n$ with $\widetilde{m} = m - (n-1)\lfloor \frac{m}{n} \rfloor$. Also, for the application of the lemma for the $i^{\text{th}}$ datapoint, we use as $U_r^{(0)}$ the $\sqrt{m}W_{(i-1)\lfloor \frac{m}{n} \rfloor + r}^{(0)}$, as $\beta_r^{(0)}$ the $\sqrt{m}b_{(i-1)\lfloor \frac{m}{n} \rfloor + r}^{(0)}$ and as $\alpha_r^{(0)}$ the $a_{(i-1)\lfloor \frac{m}{n} \rfloor + r}^{(0)}$. We apply a union bound and we have that with probability at least $1 - n\exp(-\Omega(\sqrt{m/n})) = 1 - \exp(-\Omega(\sqrt{m/n}))$, from the $n$ applications of the lemma, we get these $\Delta W^{(i)}$ and we construct $\Delta W = \left[\Delta W^{(1)}, \cdots, \Delta W^{(n)}\right] \in \mathbb{R}^{d \times m}$ and we have that

$$\|\Delta W\|_{2,\infty} \leq O\left(m^{1/3}\frac{\mathfrak{C}(q, \epsilon_3)}{\lfloor \frac{m}{n} \rfloor}\right) \leq O\left(\frac{n \, \mathfrak{C}(q, \epsilon_3)}{m^{2/3}}\right) \leq \frac{(n/\epsilon)^{O(\gamma^{-1})}}{m^{2/3}}$$

and

$$\forall x \in \mathcal{X}, \quad \left|\sum_{r=1}^{m} a_r^{(0)}\langle\Delta W_r, x\rangle \mathbb{1}\{\langle W_r^{(0)}, x\rangle + b_r^{(0)} \geq 0\} - \sum_{i=1}^{n} y_i q(\langle x_i, x\rangle)\right| \leq n\epsilon_3 \leq \epsilon/3$$

where the last inequality is a crude bound, but sufficient for our purposes. $\qquad\square$

We proceed with the proof of Lemma 10.16

*Proof.* We apply Lemma 6.6 using $\phi(z) = y_i q(z)$ and $\epsilon_1 = \epsilon_3$. Observe that since $|y_i| \leq O(1)$, the complexities of $\phi$ and $q$ are the same, up to constants. Thus, we have that there exists a function $h : \mathbb{R}^2 \to [-\mathfrak{C}(q, \epsilon_3), \mathfrak{C}(q, \epsilon_3)]$ such that

$$\forall x \in \mathcal{X}, \quad \left|\mathop{\mathbb{E}}_{u\sim\mathcal{N}(0,I_d), \beta\sim\mathcal{N}(0,1)}[\mathbb{1}\{\langle u, x\rangle + \beta \geq 0\} h(\langle x_i, u\rangle, \beta)] - y_i q(\langle x_i, x\rangle)\right| \leq \epsilon_3 \tag{20}$$

Now, we fix an $x \in \mathcal{X}$. From Hoeffding's inequality, we get that with probability at least $1 - \exp\left(-\Omega\left(\frac{\widetilde{m}\epsilon_3^2}{\mathfrak{C}^2(q,\epsilon_3)}\right)\right)$,

$$\left|\frac{1}{\widetilde{m}}\sum_{r=1}^{\widetilde{m}} \mathbb{1}\{\langle U_r^{(0)}, x\rangle + \beta_r^{(0)} \geq 0\}h\left(\langle x_i, U_r^{(0)}\rangle, \beta_r^{(0)}\right) - \mathop{\mathbb{E}}_{u\sim\mathcal{N}(0,I_d), \beta\sim\mathcal{N}(0,1)}[\mathbb{1}\{\langle u, x\rangle + \beta \geq 0\} h(\langle x_i, u\rangle, \beta)]\right| \leq \epsilon_3$$

By setting $\Delta W_r^{(i)} = \frac{1}{\alpha_r^{(0)}}\frac{2h\left(\langle x_i, U_r^{(0)}\rangle, \beta_r^{(0)}\right)}{\widetilde{m}}e_d$ (where $e_d = (0, 0, \ldots, 0, 1) \in \mathbb{R}^d$) we have that $\|\Delta W^{(i)}\|_{2,\infty} \leq O\left(m^{1/3}\frac{\mathfrak{C}(q,\epsilon_3)}{\widetilde{m}}\right)$ and since $x_d = 1/2$ for all $x \in \mathcal{X}$, we have that for every $x \in \mathcal{X}$, with probability at least $1 - \exp\left(-\Omega\left(\frac{\widetilde{m}\epsilon_3^2}{\mathfrak{C}^2(q,\epsilon_3)}\right)\right)$,

$$\left|\sum_{r=1}^{\widetilde{m}} \mathbb{1}\{\langle U_r^{(0)}, x\rangle + \beta_r^{(0)} \geq 0\}\alpha_r^{(0)}\langle\Delta W_r^{(i)}, x\rangle - \mathop{\mathbb{E}}_{u\sim\mathcal{N}(0,I_d), \beta\sim\mathcal{N}(0,1)}[\mathbb{1}\{\langle u, x\rangle + \beta \geq 0\} h(\langle x_i, u\rangle, \beta)]\right| \leq \epsilon_3$$

$$\tag{21}$$

The fact that 21 holds with overwhelming probability, enables us to take a union bound over a fine-grained net of $\mathcal{X}$. Let $c > 0$ be a sufficiently large constant (e.g. 10) and let $\mathcal{X}_1$ be a maximal $\frac{1}{\widetilde{m}^c}$-net of $\mathcal{X}$. It is well-known that $|\mathcal{X}_1| \leq \left(\frac{1}{\widetilde{m}}\right)^{O(d)}$. By applying a union bound over $\mathcal{X}_1$ for 21, we have that for $\widetilde{m} \geq c_1 \frac{d}{\epsilon_3^2} \mathfrak{C}^2(q, \epsilon_3)$ ($c_1$ is a large constant),

$$\Pr\left[\forall x \in \mathcal{X}_1, \ \left| \sum_{r=1}^{\widetilde{m}} \mathbb{1}\{\langle U_r^{(0)}, x \rangle + \beta_r^{(0)} \geq 0\} \alpha_r^{(0)} \langle \Delta W_r^{(i)}, x \rangle \right. \right. \tag{22}$$

$$\left. \left. - \underset{u \sim \mathcal{N}(0, I_d), \beta \sim \mathcal{N}(0,1)}{\mathbb{E}} \left[ \mathbb{1}\{\langle u, x \rangle + \beta \geq 0\} h(\langle x_i, u \rangle, \beta) \right] \right| > \epsilon_3 \right]$$

$$\leq \exp\left(O(d \log m)\right) \exp\left(-\Omega\left(\frac{\widetilde{m}\epsilon_3^2}{\mathfrak{C}^2(q, \epsilon_3)}\right)\right) \tag{23}$$

$$= \exp\left(-\Omega\left(\frac{\widetilde{m}\epsilon_3^2}{\mathfrak{C}^2(q, \epsilon_3)}\right)\right) \tag{24}$$

The final step is to show that with overwhelming probability, for all $x \in \mathcal{X}_1$, if we perturb $x$ by at most $\frac{1}{\widetilde{m}^c}$ in $\ell_2$, then the LHS of 21 changes very slightly. Because $c$ can be chosen to be as large constant as we want, this "stability" requirement is very mild and also straightforward to prove. We proceed with a formal proof.

We will show the stability property for a fixed $x \in \mathcal{X}$ and then we will do a union bound. Let $v \in \mathbb{R}^d$ such that $x + v \in \mathcal{X}$ and $\|v\|_2 \leq \frac{1}{\widetilde{m}^c}$. This $v$ can be arbitrarily correlated with the randomness $\{U_r^{(0)}, \beta_r^{(0)}, \alpha_r^{(0)}\}_{r=1}^{\widetilde{m}}$. We will show the following claim

**Claim 10.17.** *For all $x \in \mathcal{X}_1$, with probability at least $1 - \exp(-\Omega(\sqrt{\widetilde{m}}))$,*

$$D_1 := \left| \sum_{r=1}^{\widetilde{m}} \mathbb{1}\{\langle U_r^{(0)}, x+v \rangle + \beta_r^{(0)} \geq 0\} \alpha_r^{(0)} \langle \Delta W_r^{(i)}, x+v \rangle - \sum_{r=1}^{\widetilde{m}} \mathbb{1}\{\langle U_r^{(0)}, x \rangle + \beta_r^{(0)} \geq 0\} \alpha_r^{(0)} \langle \Delta W_r^{(i)}, x \rangle \right|$$

$$\leq O\left(\frac{\mathfrak{C}(q, \epsilon_3)}{\sqrt{\widetilde{m}}}\right)$$

*and*

$$D_2 := \left| \underset{u \sim \mathcal{N}(0, I_d), \beta \sim \mathcal{N}(0,1)}{\mathbb{E}} \left[ \mathbb{1}\{\langle u, x+v \rangle + \beta \geq 0\} h(\langle x_i, u \rangle, \beta) \right] \right.$$

$$\left. - \underset{u \sim \mathcal{N}(0, I_d), \beta \sim \mathcal{N}(0,1)}{\mathbb{E}} \left[ \mathbb{1}\{\langle u, x \rangle + \beta \geq 0\} h(\langle x_i, u \rangle, \beta) \right] \right|$$

$$\leq O\left(\frac{\mathfrak{C}(q, \epsilon_3)}{\sqrt{\widetilde{m}}}\right)$$

With this claim at hand we can finish the proof of the Lemma 10.16. Indeed, combining 20, 22 and the above claim, we have that with probability at least $1 - \exp\left(-\Omega\left(\frac{\widetilde{m}\epsilon_3^2}{\mathfrak{C}^2(q, \epsilon_3)}\right)\right) - \exp(-\Omega(\sqrt{\widetilde{m}}))$,

$$\forall x \in \mathcal{X},$$

$$\left| \sum_{r=1}^{\widetilde{m}} \alpha_r^{(0)} \langle \Delta W_r^{(i)}, x \rangle \mathbb{1}\{\langle U_r^{(0)}, x \rangle + \beta_r^{(0)} \geq 0\} - y_i q(\langle x_i, x \rangle) \right| \leq O\left(\frac{\mathfrak{C}(q, \epsilon_3)}{\sqrt{\widetilde{m}}}\right) + 2\epsilon_3$$

since $\widetilde{m} \geq c_1 \frac{d}{\epsilon_3^2} \mathfrak{C}^2(q, \epsilon_3)$ for a large constant $c_1$, we are done. $\qquad\square$

It remains to prove the Claim 10.17.

*Proof.* We start with bounding $D_1$. Observe that from the way we constructed $\Delta W^{(i)}$, we have that for $j \leq d - 1$, $\Delta W_{rj}^{(i)} = 0$. At the same time, $v_d = 0$, so $\langle \Delta W_r^{(i)}, v \rangle = 0$. Using that $\|\Delta W^{(i)}\|_{2,\infty} \leq O\left(m^{1/3} \frac{\mathfrak{C}(q, \epsilon_3)}{\widetilde{m}}\right)$ and $|\alpha_r^{(0)}| = \frac{1}{m^{1/3}}$, we get that

$$D_1 \leq O\left(\frac{\mathfrak{C}(q,\epsilon_3)}{\widetilde{m}}\right) \sum_{r=1}^{\widetilde{m}} \left| \mathbb{1}\{\langle U_r^{(0)}, x+v\rangle + \beta_r^{(0)} \geq 0\} - \mathbb{1}\{\langle U_r^{(0)}, x\rangle + \beta_r^{(0)} \geq 0\} \right|$$

$$\leq O\left(\frac{\mathfrak{C}(q,\epsilon_3)}{\widetilde{m}}\right) \sum_{r=1}^{\widetilde{m}} \mathbb{1}\left\{ \mathrm{sgn}(\langle U_r^{(0)}, x+v\rangle + \beta_r^{(0)}) \neq \mathrm{sgn}(\langle U_r^{(0)}, x\rangle + \beta_r^{(0)}) \right\}$$

$$\leq O\left(\frac{\mathfrak{C}(q,\epsilon_3)}{\widetilde{m}}\right) \sum_{r=1}^{\widetilde{m}} \left( \mathbb{1}\left\{ |\langle U_r^{(0)}, x\rangle + \beta_r^{(0)}| \leq \frac{1}{\sqrt{\widetilde{m}}} \right\} + \mathbb{1}\left\{ \|U_r^{(0)}\|_2 > c_2\sqrt{\widetilde{m}} \right\} \right).$$

where $c_2$ can be chosen to be as large as we want (but still a constant) as long as we choose the constant $c$, that appears at the construction of the net, to be sufficiently large. We prove the following claim, whose proof is almost identical to the proof of Claim 10.18, but we provide it for completeness.

**Claim 10.18.** *With probability at least $1 - \exp(-\Omega(\widetilde{m}))$, for all $r \in [\widetilde{m}]$, $\|U_r^{(0)}\|_2 \leq O(\sqrt{\widetilde{m}})$.*

*Proof.* From concentration of sum of independent Chi-Square random variables, we have that for all $r$, with probability at least $1 - \exp(-\Omega(\widetilde{m}^2/d))$, $\|U_r^{(0)}\|_2^2 \leq O(\widetilde{m})$. Since $\widetilde{m} \geq d$, a union bound over all $r$ finishes the proof of the claim. $\square$

Thus, by appropriately choosing $c_2$, we get that with probability at least $1 - \exp(-\Omega(\widetilde{m}))$,

$$D_1 \leq O\left(\frac{\mathfrak{C}(q,\epsilon_3)}{\widetilde{m}}\right) \sum_{r=1}^{\widetilde{m}} \mathbb{1}\left\{ |\langle U_r^{(0)}, x\rangle + \beta_r^{(0)}| \leq \frac{1}{\sqrt{\widetilde{m}}} \right\}$$

Now, $\mathbb{1}\left\{ |\langle U_r^{(0)}, x\rangle + \beta_r^{(0)}| \leq \frac{1}{\sqrt{\widetilde{m}}} \right\}$ are $\widetilde{m}$ independent Bernoulli random variables and because of Claim 10.1, the corresponding probability is at most $O\left(\frac{1}{\sqrt{\widetilde{m}}}\right)$. Thus, from Chernoff bounds we get that with probability at least $1 - \exp(-\Omega(\sqrt{\widetilde{m}}))$, $\sum_{r=1}^{\widetilde{m}} \mathbb{1}\left\{ |\langle U_r^{(0)}, x\rangle + \beta_r^{(0)}| \leq \frac{1}{\sqrt{\widetilde{m}}} \right\} \leq O(\sqrt{\widetilde{m}})$. By applying a union bound, we get that with probability at least $1 - \exp(-\Omega(\sqrt{\widetilde{m}})) - \exp(-\Omega(\widetilde{m})) = 1 - \exp(-\Omega(\sqrt{\widetilde{m}}))$, $D_1 \leq O\left(\frac{\mathfrak{C}(q,\epsilon_3)}{\sqrt{\widetilde{m}}}\right)$.

We proceed with bounding $D_2$. Since $|h(\cdot)| \leq \mathfrak{C}(q,\epsilon_3)$, we have

$$D_2 \leq \mathfrak{C}(q,\epsilon_3) \mathop{\mathbb{E}}_{u\sim\mathcal{N}(0,I_d),\beta\sim\mathcal{N}(0,1)} [|\mathbb{1}\{\langle u, x+v\rangle + \beta \geq 0\} - \mathbb{1}\{\langle u, x\rangle + \beta \geq 0\}|]$$

$$\leq \mathfrak{C}(q,\epsilon_3) \mathop{\mathbb{E}}_{u\sim\mathcal{N}(0,I_d),\beta\sim\mathcal{N}(0,1)} \left[ \mathbb{1}\{|\langle u, x\rangle + \beta| \leq \frac{1}{\sqrt{\widetilde{m}}}\} + \mathbb{1}\{\|u\|_2 > c_2\sqrt{\widetilde{m}}\} \right]$$

where $c_2$ is the same constant as before. But, same as before, $\Pr_{u\sim\mathcal{N}(0,I_d),\beta\sim\mathcal{N}(0,1)}\left[|\langle u, x\rangle + \beta| \leq \frac{1}{\sqrt{\widetilde{m}}}\right] \leq O(\frac{1}{\sqrt{\widetilde{m}}})$ and $\Pr_{u\sim\mathcal{N}(0,I_d),\beta\sim\mathcal{N}(0,1)}\left[\|u\|_2 > c_2\sqrt{\widetilde{m}}\right] \leq \exp(-\Omega(\widetilde{m}))$. So, $D_2 \leq O\left(\frac{\mathfrak{C}(q,\epsilon_3)}{\sqrt{\widetilde{m}}}\right)$. $\square$