[Reviews · NeurIPS 2020]

Review 1

Summary and Contributions: 1. The paper proves that for two layer neural networks, adversarial training finds a robust model near the initialization. 2. The paper also provides a new approximation theory, which shows that the step function can be approximated well by a polynomially wide ReLU network whose weights are close to the initialization.

Strengths: 1. The question of how over-parameterized a network should be, so that it is adversarially robust, is really important. Answering this question will help in the design of robust networks. This work takes a step in that direction and improves upon some aspects of previous work. 2. The proof of the claims seem correct and are really well explained.

Weaknesses: 1. The $\gamma$-separability requirement seems odd in the sense that it also requires samples from the same class to be separated by a margin. Earlier work by Gao et al.(2019) does not seem to have this requirement. This seems artificial and probably the analysis can be tightened to fix this. Would replacing such close samples by just one 'representative' sample from the same class help? 2. The previous work by Gao et al.(2019) and this work, both only guarantee robustness against the adversary used during training. Can something be said about the robustness against the most intelligent adversary $\mathcal{A}^*$, for a network trained using a polynomial time adversary $\mathcal{A}$? This seems to be much more important question than whether a network trained using a particular adversary can defend against the same adversary. ------ Post author feedback comments ------- My concerns have been satisfactorily answered in the feedback.

Correctness: The proof overview (section 5) and proof of Theorem 5.3 (section 6) seems to be correct and are well explained.

Clarity: The paper is really well written. It was easy to understand the intuitions and the flow of the proof.

Relation to Prior Work: The relation to prior work has been discussed well and differences have been highlighted.

Reproducibility: Yes

Additional Feedback: Please see the Weakness section above.


Review 2

Summary and Contributions: This paper follows a recent line of work that analyzes the convergence properties of adversarial training. The authors show that adversarial training always converges in polynomial time on two-layer ReLU networks assuming the separability of training data. The general idea is to construct a robust net near initialization using polynomial approximation.

Strengths: This paper is of interesting topic, clear problem formulation and important results. The authors makes a detailed analysis and proof of the convergence properties of adversarial training in polynomial time on the two-layer ReLU network by using polynomial approximation, which significantly improves the previous theoretical results. The logic of the whole paper is clear and the proof part is easy to understand. The convergence property of standard adversarial training is clearly revealed. The proof in the supplementary material is detailed and the conclusions are clear. In addition, the new results of approximation theory can be further applied to the theory of over-parameterized model.

Weaknesses: I don’t have much negative observations about this paper.

Correctness: There is no obvious error as far as I can see.

Clarity: The full paper is clear and easy to follow.

Relation to Prior Work: The authors introduce adversarial example and defense, convergence of adversarial training and polynomial approximation in the related work part. I think the introduction to the prior work is sufficient.

Reproducibility: Yes

Additional Feedback: The paper could benefit from presenting the principle of adversarial attack in the related-work section. In addition, although the main contribution of this paper lies in the theoretical analysis of the convergent properties of the adversarial training, simple experiments on some small datasets may further verify the conclusions of this paper and demonstrate the correctness of the conclusions more intuitively. ========================== After rebuttal: Thanks author for putting the updated results. It does solve some concerns to me. This is a promising submission and I would maintain my original score of 7.


Review 3

Summary and Contributions: The paper studies the convergence theory of over-parameterized adversarial training. Based on Wang et al.'s work, it achieves further results: it proves the convergence to low robust training loss of two-layer ReLU activated neural network in standard adversarial training and for polynomial width and running time of the input dimension d.

Strengths: The paper gives a proof overview of the main result Theorem 4.1, so it is easy to understand. The assumption, gamma-separability is very clever. It helps the proof a lot. Also, its rationality is verfied empirically in popular dataset. Settings are more realistic. It solve the curse of dimensionality (which is a future work of Wang et al.'s work) perfectly.

Weaknesses: Although this paper has an obvious improvement on Gao et al.'s work, I have to say that it lacks novelty, and the contribution is small. Width, runing time and activation funtion are not a huge gap in Gao et al.'s work. What I really want to see is to remove projection in deep net, improve the online learning or analysis the robust generalization. However, both the results and the proof methods have little inspiration. This paper claims that '(Gao et al.) require the width of the net and the running time to be exponential in input dimension d, and they consider an activation function that is not used in practice' in abstract. This is likely to cause a serious misleading. In fact, the main results of Gao et al.'s work, Theorem 5.1 and 5.2 only requires polynomial of d with a seldom used activation function. However, it is quadratic ReLU activation, which is a common activation function, that the width and running time are exponential with. Gao et al. put the exponential case in appendix C.2 (instead of C.1, this paper makes a misleading mistake in section 1) in order not to use the Lipschitz assumption. These words may make readers think Wang et al. require both exponential width and seldom used activation. ---------------------------------------------------------- Post rebuttal: The authors' rebuttal solves some of concerns, and I would raise my score to weak accept.

Correctness: I have carefully checked the results and proofs of this paper, and ensure that it is correct.

Clarity: Its writing is fluent and there are no obscure places. The structure is clear and concise. It also shows some important proof technique in advance.

Relation to Prior Work: I am afraid not. Although authors show they have a good understanding of prior work's (especially Wang et al.'s work) contributions and defects between the lines, as said above, the expression in some places may leads a serious mistake.

Reproducibility: Yes

Additional Feedback: The assumption, gamma-seperable can be verified on a large dataset, such as Imagenet. It would be better if it could give a stronger intuitive feeling.


Review 4

Summary and Contributions: The paper presents a new theoretical result on adversarial training of two-layer ReLU networks, more concretely, it shows that for networks of polynomial width the adversarial training algorithm converges to arbitrarily small robust training loss.

Strengths: I find the result interesting and an important step to understanding adversarial training, since the conditions hold in practical scenarios (e.g., CIFAR10) and for networks of polynomial widths (as opposed to exponentially wide networks, considered before).

Weaknesses: While the main focus of paper is theoretical, I feel like it could benefit from small toy examples experimentally demonstrating the authors' claim. E.g., plot the dependence of robust training loss vs depth.

Correctness: Correct.

Clarity: The paper is well written. I suggest the authors to add some explanation of the logic behind the definition 6.1. Additionally, some visualizations (as noted before) could simplify the understanding. For instance, a visualization explaining definition 3.4 could be helpful.

Relation to Prior Work: Related work is thoroughly reviewed.

Reproducibility: Yes

Additional Feedback: Please see the previous comments.

[Author Response · NeurIPS 2020]

We thank all the reviewers for their positive and useful feedbacks, which we will use to improve the paper. We first address the common comments from reviewers:

## To Reviewer #1

**@ The $\gamma$-seperatability requirement is odd since it also requires samples from the same class to be separated:** Good point! Actually, $\gamma$-seperatability can be replaced by separation only among different classes. With minor modifications, our results can be applied to classification losses, like cross-entropy. For such losses, we can define $\delta=$ minimum distance between different classes and thus $1/\gamma$ not being large becomes even more practically relevant, as an assumption (remember that $\gamma = \delta(\delta - 2\rho)$). We will add more discussions in the next version.

**@ Can something be said about the robustness against the most intelligent adversary for a network trained using a polynomial time adversary?:** This is certainly an interesting future direction. This might require characterization of what polynomial time adversary means, since training with a trivial adversarial which only generates the original data doesn't lead to robustness against the most intelligent adversary.

Note that our results apply to any adversary (including the most intelligent one) used in training.

## To Reviewer #2

We appreciate the suggestions and will update the related-work section accordingly. Regarding experiments, we note previous works (e.g. Figure 4 in Madry et al.) have already shown empirically that reasonably wide networks achieve small robust training error. Our work serves as a potential theoretical explanation for this phenomenon.

## To Reviewer #3

**@ "Wang et al.":** We are unable to find any related works with the reference "Wang et al". Given the comments, we believe the reviewer meant to refer to Gao et al.

**@ the paper lacks novelty, and the contribution is small because Gao et al. only requires poly(d) width with a seldom used activation function where d is the input dimension:** This claim made by the reviewer is incorrect. In Gao et al., in order to achieve small robust training loss, Theorem 5.2 and Corollary 5.1 require the width to be polynomial in the constant $R_{D,B,\epsilon}$, and Gao et al did not to show $R_{D,B,\epsilon}$ is poly(d) with *any* activation function. In fact, they only managed to upper bound $R_{D,B,\epsilon}$ by $(1/\epsilon)^d$ with the quadratic ReLU activation. One of our main contribution is to bound $R_{D,B,\epsilon}$ by poly(d) with the ReLU activation using a novel analysis.

This misunderstanding is possibly due to a claim made in Gao et al.'s intro that "we show that projected gradient descent converges to a network where the surrogate loss with respect to the attack $\mathcal{A}$ is within $\epsilon$ of the optimal robust loss. The required width is polynomial in the depth and the input dimension." We note that although this claim is correct, the optimal robust loss in their setting may not be necessarily small. The only concrete case where they prove it is small is for quadratic ReLU networks with $(1/\epsilon)^d$ width (Theorem C.1 in Gao et al.).

**@ it is misleading that the authors claim quadratic ReLU is not used in practice:** We are not aware of the use of quadratic ReLU in any practical setting but are happy to change that phrase if the reviewer could give us some references. But it is important to note that even for the quadratic ReLU, the upper bound in Gao et al. is exponential in d. That is the main point in that line.

**@ appendix C.2 instead of C.1** : We thank the reviewer for pointing out this typo. We meant to write theorem C.1. We will fix it in the next version.

## To Reviewer #4

**@ the paper could benefit from small toy examples experimentally demonstrating the authors' claim. E.g., plot the dependence of robust training loss vs depth:** By "loss vs depth", did the reviewer mean loss vs width? If so, many previous works (e.g. Figure 4 in Madry et al.) have already showcased the suggested experiments. They show larger width leads to lower robust training loss. Our work serves as a theoretical explanation for such experiments.

[Meta-Review · NeurIPS 2020]

This paper proves that adversarial training of over-parameterized neural networks converges to a robust solution. Specifically, the paper studies two-layer ReLU networks with width that is polynomial in the input dimension, d, the number of training points, n, and the inverse of the robustness parameter, 1/\epsilon. (This improves over prior work that required exponentially wide networks.) The proof is by construction; an algorithm is proposed that, in poly(d, n, 1/\epsilon) iterations, finds a network with poly(d, n, 1/\epsilon) width that is \epsilon-robust. Adversarial training is an important and rapidly expanding field of ML. This paper fills in some gaps w.r.t. over-parameterized neural networks (which, on their own, have recently created a cottage industry theory papers). The reviewers seem to really appreciate the paper, saying that it solves a relevant problem, and that it's clear and well written. Importantly, the authors' response appears to have cleared up a few misunderstandings from the first round of reviews. I thank the authors for writing a succinct, helpful response; and I thank the reviewers, for carefully considering what the authors wrote and incorporating this feedback into their reviews. This is a good example of the author response process working as it should. I encourage the authors' to incorporate feedback from the reviewers when revising the paper.